# Spontaneous neural synchrony links intrinsic spinal sensory and motor networks during unconsciousness

Jacob Graves McPherson[1,2,3,4]*, Maria F Bandres[1,5]

[1]Program in Physical Therapy, Washington University School of Medicine, St. Louis, United States; [2]Department of Anesthesiology, Washington University School of Medicine, St. Louis, United States; [3]Washington University Pain Center, Washington University School of Medicine, St. Louis, United States; [4]Program in Neurosciences, Washington University School of Medicine, St. Louis, United States; [5]Department of Biomedical Engineering, Washington University School of Medicine, St. Louis, United States

**Abstract** Non-random functional connectivity during unconsciousness is a defining feature of supraspinal networks. However, its generalizability to intrinsic spinal networks remains incompletely understood. Previously, Barry et al., 2014 used fMRI to reveal bilateral resting state functional connectivity within sensory-dominant and, separately, motor-dominant regions of the spinal cord. Here, we record spike trains from large populations of spinal interneurons in vivo in rats and demonstrate that spontaneous functional connectivity also links sensory- and motor-dominant regions during unconsciousness. The spatiotemporal patterns of connectivity could not be explained by latent afferent activity or by populations of interconnected neurons spiking randomly. We also document connection latencies compatible with mono- and disynaptic interactions and putative excitatory and inhibitory connections. The observed activity is consistent with the hypothesis that salient, experience-dependent patterns of neural transmission introduced during behavior or by injury/disease are reactivated during unconsciousness. Such a spinal replay mechanism could shape circuit-level connectivity and ultimately behavior.

*For correspondence:
mcpherson.jacob@wustl.edu

## Introduction

Synchronous neural activity across functionally and spatially distinct brain structures, that is, functional connectivity, is a hallmark of sensorimotor integration, cognition, and behavior during periods of attentive wakefulness. Recent elucidation of brain networks intrinsically active during unconsciousness and inattentive wakefulness has led to a substantially more nuanced view of brain function (*Demertzi et al., 2019*; *Fox et al., 2005*; *Greicius et al., 2003*; *Mashour and Hudetz, 2018*; *Raichle et al., 2001*; *Steriade et al., 1993*; *Wenzel et al., 2019*). Unconscious network activity spans multiple spatiotemporal scales and has known functions ranging from circuit-level synaptic stabilization (*Puentes-Mestril and Aton, 2017*; *Tsodyks et al., 1999*; *Wei et al., 2016*) to maintenance of ongoing physiological processes (*Sanchez-Vives et al., 2017*). Although the finding of spontaneous, non-random network activity during unconsciousness appears to be robust across different functional regions of the brain, it has yet to be unequivocally confirmed whether this phenomenon is a conserved feature of complex neural systems that generalizes to the spinal cord.

Patterns of resting state functional connectivity in the spinal cord have only been preliminarily characterized (*Barry et al., 2014*; *Chen et al., 2015*; *Conrad et al., 2018*; *Eippert et al., 2017*; *Kong et al., 2014*; *Tl et al., 2019*). The most reliable findings to date have been correlations between spontaneous BOLD signals in the left and right dorsal horns, and, separately, the left and

right ventral horns (VHs) (*Barry et al., 2014*; *Eippert et al., 2017*; *Kong et al., 2014*; *Tl et al., 2019*). Spontaneous connectivity between the dorsal horn and VH, between the intermediate gray (IG) and the VH, and within the VH itself has yet to be reliably delineated.

Other gaps also exist. For example, it is unknown whether network topologies evinced by spinal BOLD signals mirror those drawn from spike trains of individual neurons. Indeed, BOLD signals are only indirectly linked to spiking activity (*Logothetis et al., 2001*; *Murayama et al., 2010*; *Vakorin et al., 2007*), which is compounded by the relatively coarse spatiotemporal resolution of fMRI in the spinal cord. It is also not readily apparent whether structured activity at the single-unit level actually persists in spinal networks during unconsciousness in the absence of evoked neural transmission. The most relevant evidence, which suggests that aggregate multi-unit and local field potential activity in the dorsal horn is broadly correlated with dorsal horn BOLD fluctuations, was made during mechanical probing of the dermatome (*Tl et al., 2019*).

The potential function(s) of resting state intraspinal connectivity are likewise unknown. An intriguing possibility is that it plays a role in adaptive or maladaptive neural plasticity through a form of reactivation and synaptic stabilization during unconsciousness. This hypothesis is drawn from the function of supraspinal network activity during sleep (*Abel et al., 2013*; *Puentes-Mestril and Aton, 2017*; *Wei et al., 2016*) and is supported by the finding of altered patterns of BOLD-based intraspinal functional connectivity in conditions associated with maladaptive neural plasticity in spinal networks (*Chen et al., 2015*; *Conrad et al., 2018*). To have a direct role in shaping neural plasticity, however, a necessary substrate would be the tandem presence of synchronous discharge amongst populations of individual units spanning multiple spatial and functional regions.

Given the critical role played by the spinal cord in sensorimotor integration (broadly) and reflexes (specifically), we reasoned that spontaneous functional connectivity between neurons in sensory-dominant and motor-dominant regions of the gray matter would be a precondition for functional network activity during unconsciousness, regardless of its function. And for the reasons noted above, such a finding would have important implications for both the physiological and pathophysiological states. Several fundamental questions remain unresolved, however. Here, we address three. First, is neuron-level functional connectivity evident in regions of the spinal gray matter not traditionally associated with primary afferent inflow? Second, is spontaneous functional connectivity evident between sensory and motor regions of the gray matter? And third, does the proportion of spontaneously active neurons exhibiting correlated discharge, as well as their topology, depart from that which would be expected amongst an interconnected population of statistically similar neurons firing uncooperatively (i.e., randomly)?

We addressed these questions in vivo in rats, recording large populations of single units throughout the dorsoventral extent of the lumbar enlargement. We find that robust spontaneous neural activity is prevalent throughout the gray matter during unconsciousness and that neurons in sensory and motor regions exhibit significant, non-random correlations in their spatiotemporal discharge patterns. We also find a substantial portion of connection latencies consistent with mono- and disynaptic interactions, offering clues to a possible mechanism by which intrinsic network activity could directly shape synaptic plasticity.

## Materials and methods

All experiments were approved by the Institutional Animal Care and Usage Committees at Florida International University and Washington University in St. Louis.

### Surgical procedures, electrode implantation

Experiments were performed in adult male Sprague–Dawley rats ($N$ = 24; weight), divided across two cohorts. Fifteen animals received urethane anesthesia (1.2 g/kg i.p.). The remaining nine animals received inhaled isoflurane anesthesia (2–4% in $O_2$). Heart rate, respiration rate, body temperature, and $SpO_2$ were monitored continuously during the experiments (Kent Scientific, Inc), and temperature was regulated via controlled heating pads.

In a terminal, aseptic procedure, a skin incision was made over the dorsal surface of the T1–S1 vertebrae and the exposed subcutaneous tissue and musculature were retracted. The T13–L3 vertebrae were cleaned of musculotendonous attachments using a microcurette and the vertebral laminae were removed to expose spinal segments L4–6. The rat and surgical field were then transferred to an anti-vibration air table (Kinetic Systems, Inc) enclosed in a dedicated Faraday cage.

Clamps were secured to the vertebrae rostral and caudal to the laminectomy site, and the rat's abdomen was elevated such that respiration cycles did not result in upward or downward movement of the chest cavity or spinal cord. Under a surgical microscope (Leica Microsystems, Inc), the exposed spinal meninges were incised rostrocaudally and reflected. The spinal cord was then covered in homeothermic physiological ringer solution.

A custom four-axis motorized micromanipulator with submicron resolution (Siskiyou Corp.) was then coarsely centered over the laminectomy site. A silicon microelectrode array (NeuroNexus, Inc) custom electrodeposited with activated platinum-iridium electrode contacts (Platinum Group Coatings, Inc) was mated via Omnetics nano connectors to a Ripple Nano2+ Stim headstage (Ripple Neuro, Inc). The microelectrode array contained two shanks, each with 16 individual electrode contacts spaced uniformly at 100 µm intervals (*Figure 1a*). Electrode impedance ranged from ~1 to –4 kΩ per contact. The headstage was then securely fastened to the micromanipulator for implantation. During implantation, the data acquisition system was configured for online visualization of multi-unit and spiking activity from all 32 electrodes. Neural waveforms for specific electrode channels were also patched into an audio monitor (A-M Systems, Inc) for additional real-time feedback.

The electrode implantation site targeted the tibial branch of the sciatic nerve, with particular emphasis on sensitivity to receptive fields on the glabrous skin of the plantar surface of the ipsilateral hindpaw toes. The implantation site corresponded closely to the L5 spinal nerve dorsal root entry zone in all animals. Initial implantation site verification was performed by mechanically probing the L5 dermatome, specifically on the plantar aspect of the ipsilateral hindpaw, with the bottom-most electrodes of the microelectrode array being in contact with the dorsal roots at their entry zone. If clearly correlated multi-unit neural activity was evident, the probe was slowly advanced ventrally in

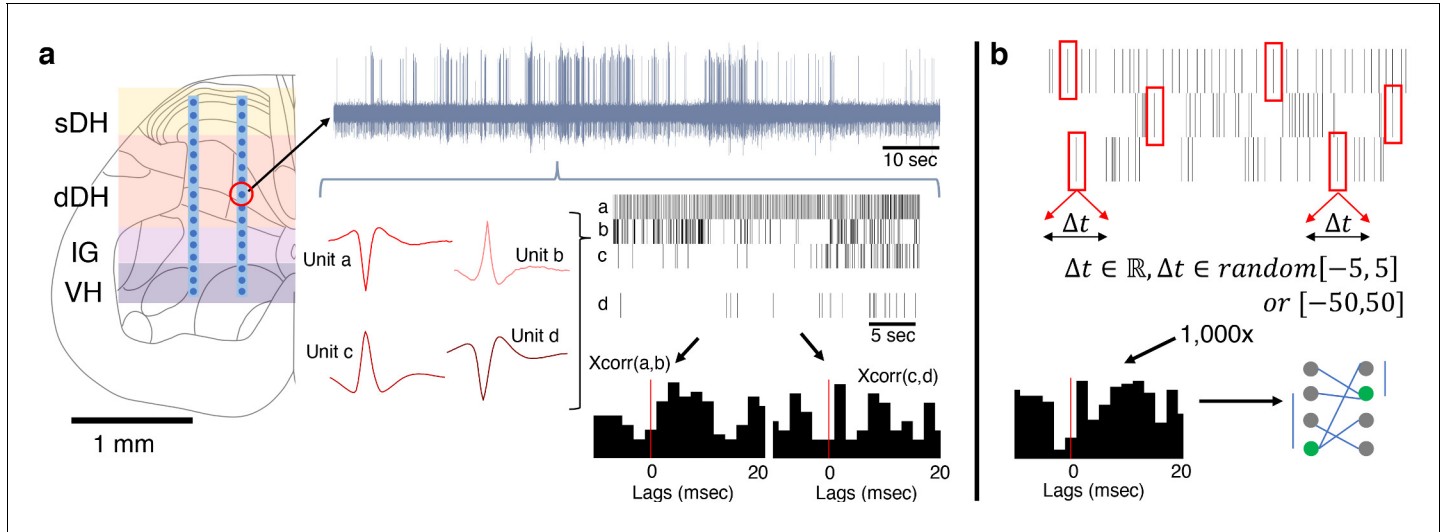

**Figure 1.** Experimental setup and design. (**a**) Dual-shank microelectrode arrays with 32 independent recording contacts were implanted into the spinal cord at the L5 dorsal root entry zone. Electrodes spanned the superficial dorsal horn (sDH), deep dorsal horn (dDH), intermediate gray matter (IG), and the ventral horn (VH). Multi-unit neural activity was recorded from each electrode (e.g., upper gray trace) and discriminated offline into spike trains of individual units (red single-unit waveforms and spike train raster plots depict four units found on a single channel). Temporal synchrony between spontaneously co-active units was then analyzed via correlation-based approaches (cross-correlation 'xcorr' histograms below rasters, vertical red lines illustrate the 0 ms lag point). (**b**) Illustration of procedure for generating the synthetic dataset. Each spike, from every identified neuron in every trial, was randomly jittered by [−5:5] ms or, separately, [−50:50] ms. The jittered data were then reconstructed, forming synthetic trials containing neurons with firing properties that were statistically similar to the observed data. This process was then repeated over 1000× to generate a large synthetic dataset from which to sample. Spatiotemporal correlation analyses then proceeded on this synthetic dataset to benchmark the empirically observed data.

25 µm increments until the deepest row of electrodes was ~200 µm deep to the dorsal surface of the spinal cord. The L5 dermatome was again probed to verify alignment between neural activity at the implantation site and the dermatome. If correlated multi-unit activity was again observed, the electrode continued to be advanced ventrally in 25 µm increments until the ventral-most row of electrode contacts was 1600–1800 µm deep to the dorsal surface (and correspondingly, the dorsal-most row of electrode contacts, i.e., the most superficial, was 100–200 µm deep to the dorsal surface of the spinal cord).

In cases where multi-unit dorsal root activity was *not* clearly correlated with the desired hindpaw receptive field, but rather was correlated with a different receptive field (e.g., on the hairy skin of the leg), the electrode was repositioned prior to implantation. In cases where *no* discernible correlation could be observed between a receptive field and dorsal root activity, yet the electrode was positioned over the L5 dorsal root entry zone, the electrode was advanced in 25 µm increments to a depth of 200 µm ventrally into the spinal cord and the receptive field mapping procedures was performed again. If appropriate activity was observed, the electrode was tracked fully; if not, it was removed and a new track was made.

In all cases, electrodes were advanced slowly to the target depth to avoid compression of the spinal cord and minimize intraspinal trauma from shear. After every ~100–200 µm of penetration, electrode advancement was paused momentarily. Penetration was resumed when neural activity (evinced by multi-unit and spiking data from implanted channels) stabilized.

In two animals, the sciatic nerve of the ipsilateral hindlimb was exposed proximal to its bifurcation into the tibial and peroneal nerves and a silver hook electrode (A-M Systems, Inc) placed around the nerve to record electroneurographic (ENG) activity. Upon completion of surgical procedures and data collection, all animals were humanely euthanized in accordance with AVMA guidelines via overdose of sodium pentobarbital (i.p. injection of Fatal Plus solution).

## Experimental procedure

We established resting motor threshold for each animal prior to recording spontaneous neural transmission. We delivered single pulses of charge-balanced current (cathode leading, 200 µs/phase, 0 s inter-phase interval) to electrodes located in the VH, with current intensity increasing in increments of 5 µA until a muscle twitch was detected in the L5 myotome (toe twitch on ipsilateral hindpaw). Current intensity was then reduced in 1 µA steps until the twitch was undetectable. Subsequently, we increased current intensity again in 1 µA increments until a twitch was recovered. The lowest current at which a twitch was detected, across all electrodes, was considered to be resting motor threshold.

We recorded 10–20 trials of spontaneous intraspinal neural transmission per animal. Each trial lasted for ~2–5 min. Raw, broadband neural activity was sampled continuously from the microelectrode array at 30 kHz. Electrical line noise and harmonics were removed via hardware filters prior to digitization. During data acquisition epochs, data from all 32 electrode channels was streamed in real time to a 60′ flat screen monitor. These data were high-pass filtered at 750 Hz to reveal multi-unit neural activity (e.g., *Figure 1a*). On channels in which single-unit activity was readily observable, dual-window time-amplitude discriminators were used to discriminate and visualize real-time single-unit spiking activity. Prior to each trial, the dermatome was mechanically probed to ensure ongoing consistency between electrode placement and receptive field location and to assess qualitatively the overall degree of neural excitability. The latter assessment in particular was used in conjunction with vital and other physiological signs to control depth of anesthesia and ensure that neural excitability did not become progressively depressed during the data acquisition session.

For sciatic nerve recordings, we first collected trials of spontaneous, baseline ENG (~1–5 min per trial). We then induced sensory transmission in the nerve by mechanically stimulating the L4, L5, and L6 dermatomes. Specifically, we recorded ENG during periods of innocuous cutaneous stimulation of the glabrous and hairy skin and during periods of proprioceptive stimulation. Proprioceptive stimuli included ankle plantarflexion and dorsiflexion, abduction and adduction of the toes, and holding joints in a flexed or extended position for a prolonged period of time. Sensory stimuli lasted ~30 s each, with 30 s to 1 min elapsing between stimuli. Subsequently, we blocked transmission in the nerve via epineurial injection of lidocaine (20 µL, 2%) (*Gokin et al., 2001*; *Kau et al., 2006*; *Thalhammer et al., 1995*) and repeated the sensory transmission experiments described above. This pharmacological nerve block is a form of deafferentation that avoids the confound of ectopic

discharge sometimes observed with mechanical sectioning of the nerve. ENG was sampled at 30 kHz (Ripple Neuro, Inc), and filtered offline to remove electrical noise (60, 120, and 180 Hz). We applied a broadband filter to the data (40 Hz to 15 kHz) to enable detection of compound action potentials/multi-unit activity as well as any potential single-unit action potentials.

## Discrimination of units, correlation and functional connectivity analyses

Single-unit neural activity was discriminated offline using the unsupervised, wavelet-based clustering approach 'wave_clus' (parameters: bandpass filter: 1 Hz to 15 kHz; minimum detection threshold: 4 standard deviations [SD] from mean; maximum detection threshold: 25 SD; detection thresholds on both positive and negative deviations; filter order for detection: 4; filter order for sorting: 2) (*Quiroga et al., 2004*). The veracity of discriminated units was verified manually. Spurious and/or duplicative units were identified and eliminated, with particular focus on units discriminated on the same or adjacent electrodes. Exclusion criteria were both quantitative (e.g., predominance of ISI < 2 ms) and qualitative (e.g., non-physiological shape, inappropriate action potential duration). Functional connectivity analyses then proceeded as follows on a per-trial basis, where pairs of units found to exhibit statistically significant temporal synchrony were deemed 'functionally connected.'

First, we computed the cross-correlation of all unique pairs of admissible units from the 32-channel microelectrode array, effectively analogous to computing peri-spike time histograms for each pair (*Figure 1a*). These computations were performed without regard to the anatomical/spatial location of the units and without defining each units of a pair as either pre- or post-synaptic. Connection latency was taken to be the time to peak correlation strength. Connection polarity (excitatory or inhibitory) was inferred using the normalized cross-correlation approach (*Pastore et al., 2018*; *Shao and Chen, 1987*).

We then quantified the strength of correlation by adapting an approach originally developed to be compatible with spike trains containing a relatively small numbers of spikes (*Gerstein and Aertsen, 1985*; *Shao and Tsau, 1996*). This calculation led to a correlation coefficient analogous to the Pearson correlation coefficient common in linear regression. If the number of spikes per train is sufficiently low ($N \leq \sim 50$), it is possible to use this approach to compute p-values via Fisher's exact test (*Shao and Tsau, 1996*). However, our surprisingly vigorous spontaneous neural transmission (see Results), coupled with the length of each trial, rendered Fisher's exact test largely intractable. As the number of spikes in a train increases, however, the distribution of spike times approximates the chi-square distribution and enables that statistic and associated degrees of freedom be used for computation of p-values associated with each correlation coefficient.

Given the large number of neurons discriminated per trial (~50–70 on average), and thus the large number of unit-pair combinations in which we computed correlation strength, careful attention was paid to multiple comparison corrections to minimize the prevalence of falsely concluding that a pair of units was significantly correlated. Controlling the family-wise error rate by applying Bonferroni correction to each test, as is often used for post-hoc multiple comparisons corrections in statistical inference, is inappropriate for datasets such as ours with trials containing extremely large numbers of non-independent comparisons (*Shao and Tsau, 1996*). Therefore, we instead used the Benjamini–Hochberg procedure to control the false discovery rate of our data on a per-trial basis. This approach ensures that the proportion of false-positive findings amongst all findings deemed to be significant is no more than specified level (in our case, 5%). The Benjamini–Hochberg procedure is applied at the trial level, and the specific p-value deemed to indicate statistical significance is a function of the data from which the statistics are being inferred. Thus, the significant p-value may be relatively more or less across different trials. Controlling the false discovery rate is a validated method for multiple comparisons corrections with datasets containing large numbers of comparisons, and it is particularly effective for situations in which certain elements being compared in a trial are likely to be more or less correlated than others due to factors such as anatomical connectivity (e.g., voxelwise comparisons of fMRI data, where distance between voxels may influence correlation strength based on the anatomy/structure-function relationships of the sampled neural structures) (*Lindquist and Mejia, 2015*).

To characterize topological aspects of functional connectivity, we classified the significantly correlated unit pairs based on their gross anatomical locations as well as the electrode from which their correlated units were discriminated. Gross anatomical locations included the superficial dorsal horn (sDH), ranging from the dorsal surface of the spinal cord to ~400 µm in depth and corresponding

approximately to Rexed's Laminae I–III; the deep dorsal horn (dDH), ranging from ~500 to 1000 μm and corresponding approximately to Rexed's Laminae III/IV–VI; the IG, ranging from ~1100 to 1300 μm, corresponding to Rexed's Laminae VII–VIII; and the VH, ranging from ~1400 to 1600+ μm and including Rexed's Laminae VIII–IX. We define the 'most connected nodes' for a given trial as the electrodes containing a significantly greater number of significant unit-pair connections than the mean number of connections across all electrodes in the microelectrode array.

## Synthetic data

We generated large synthetic datasets that matched the broad statistical properties of our observed data to use as an additional means of comparison and analyses (*Figure 1b*). The details of our approach to creating this synthetic dataset have been described previously (*Amarasingham et al., 2012*; *Fujisawa et al., 2008*). Briefly, however, the procedure is as follows. After performing spike sorting on each electrode channel for a given trial, we arrived at $N$ spike trains (where $N$ corresponds to the number of units discriminated during the spike sorting process for that trial). We then randomly selected a single number from one of two uniform distributions (see next paragraph) and added that value to the first spike time for, say, Unit 1. For example, if the first spike of Unit 1 occurred exactly 1 s after recording commenced, and we randomly drew a value of +3 ms, we would restate the first spike time for Unit 1 as occurring at 1.003 s after recording began. We then randomly drew another number from the same distribution (numbers were replaced after each draw) and added that value to the second spike time of Unit 1. This process continued for all spike times for Unit 1 during the trial. We refer to this process as jittering. The same process was subsequently performed for *all units* discriminated in that trial, resulting in $N$ jittered spike trains. After jittering all spike times for all units for a single trial, we arrived at a 'synthetic' trial. We then re-ran the correlation analyses on the jittered spike trains in the synthetic trial. By repeating this process 1000× for each unit, trial, and rat, we developed a large synthetic dataset from which statistical confidence intervals could be derived and hypothesis testing could be performed.

We created two synthetic datasets, each designed to test a different aspect of connectivity. The first synthetic dataset was designed to test short-latency connectivity, as would be observed with mono-, di-, and minimally polysynaptic connections. For this dataset, we jittered the real data using a distribution of ±[0, 1, 2, 3, 4, or 5] ms. These values simulate perturbations to short-latency interactions while preserving each unit's firing rate at a broad timescale. The second synthetic dataset was designed to test latencies compatible with complex, highly polysynaptic interactions, and used a jittering distribution of ±50 ms (also with 1 ms granularity; i.e., draw a random number from [−50,−49, −48, . . . 48, 49, 50] ms). The overall number of spikes per unit was not changed in either jittering procedure so as to avoid confounds in the interpretation of our correlation results.

## Statistical methods

Statistical inference beyond that required for the determination of significant temporal connections between pairs of co-active units (described above) is largely based on analysis of variance (ANOVA) techniques for both the urethane and isoflurane cohorts. The normality of each dataset was confirmed prior to performing ANOVAs. For within-cohort comparisons, a main effect of anatomical region on the mean number of units, proportion of significant connections, or proportion of most connected nodes (respectively) was inferred using one-way repeated measures ANOVA formulations. Assessment of the potential significance of anatomical region (within-subjects factor), anesthetic (between-subjects factor), and their interaction on the proportion of excitatory and inhibitory connections was conducted using a two-way repeated measures ANOVA design. If data violated the assumption of sphericity, Greenhouse–Geisser correction was applied. The family-wise error rate of post-hoc testing was controlled through Bonferroni correction for all comparisons. Student's t-tests were used to determine differences between individual (non-repeated) factors. This included comparisons of the proportion of within-region vs. between-region connections for a given cohort, comparisons of the mean number of units discriminated per animal between the cohorts, and excitatory vs. inhibitory latencies for a given cohort. For both ANOVA-based and t-test-based analyses, comparisons were considered significant at the $\alpha = 0.05$ level. Data are presented in text as mean ± standard error unless otherwise noted. All statistical tests were performed in the IBM SPSS environment.

## Results

### Vigorous spontaneous activity in single units remains evident throughout sensory and motor regions of the spinal gray matter during unconsciousness

We focus on urethane anesthetized animals because urethane potently suppresses spontaneous discharge in the dorsal roots (minimizing undue afferent activity) while only modestly impacting resting membrane potential, GABA-ergic, and excitatory amino acid transmission (*Daló and Hackman, 2013*; *Hara and Harris, 2002*). Thus, urethane enables characterization of the spinal cord in a state more representative of physiological activity than many other anesthetic agents.

First, we quantified the gross anatomical distribution of spontaneously active units. In total, we recorded from approximately 860 well-isolated units across 13 urethane-anesthetized rats, averaging 66 ± 8 units per trial (e.g., *Figure 1a*). This per-animal average number of units corresponds to approximately two units discriminated per electrode across the array, with a range from 0 to ~5 units per electrode. These findings are consistent with the initial validation data for the wave_clus package, which was developed with three neurons per electrode as the benchmark (*Quiroga et al., 2004*). They are also consistent with subsequent studies using wave_clus with similar electrodes to ours, which show that 1–4 units are typically discriminated and that 5–6 units can be discriminated effectively without missed clusters or false positives (*Pedreira et al., 2012*; *Rey et al., 2015*).

A representative raster plot from one trial is shown in *Figure 2a*. Spontaneously active units can be observed throughout the dorsoventral extent of the sampled region. Broadly distributed, robust discharge was a consistent feature of all animals. Across the urethane cohort, the mean number of spontaneously active units discriminated per gross anatomical region per trial was: sDH: 11 ± 3; dDH: 25 ± 3; IG: 16 ± 2; VH: 14 ± 2 (*Figure 2b*). We found a significant main effect of region on connection number (F = 6.368, p=0.001), which was driven by a significantly greater number of units in the dDH than the sDH or VH. No other regions differed from one another (*Supplementary file 1*, tab 1a).

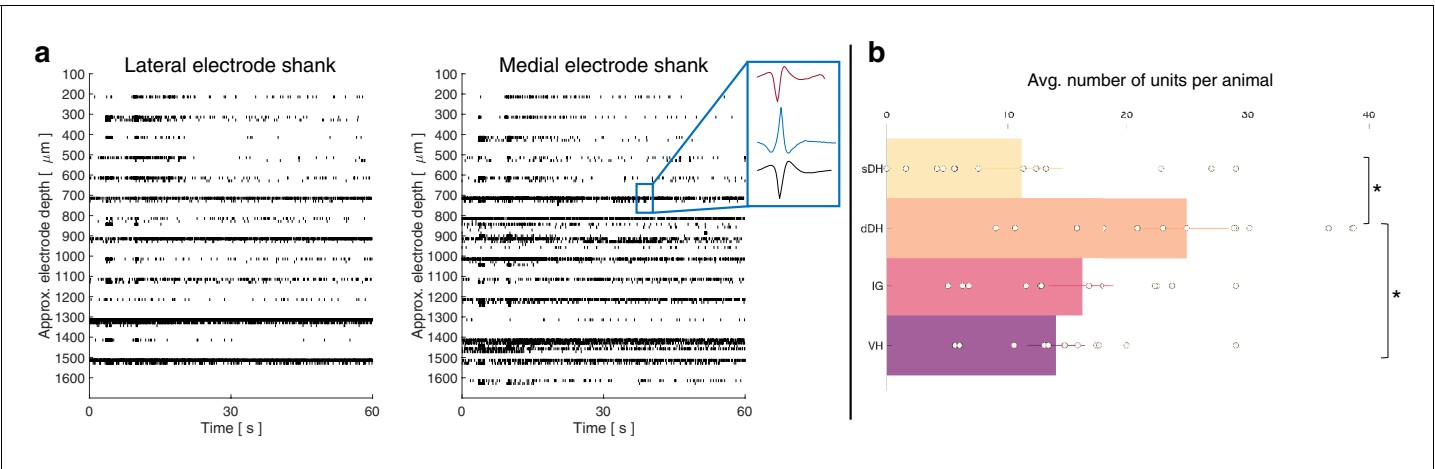

**Figure 2.** Spontaneous neural transmission is broadly evident across all spatial and functional regions of the spinal gray matter. (a) Raster plot of spontaneously active neurons. Each row of hatches represents a discrete neuron. Inset depicts representative spike waveforms discriminated from a single electrode. X-axes (time) are synchronized across the two subplots. (b) Distribution of spontaneously active units per gross anatomical region across animals in the urethane cohort (μ ± sem; N = 13 animals). The deep dorsal horn contained significantly more spontaneously active units on average than the superficial dorsal horn or ventral horn, driving an overall main effect of region (p=0.001).

# Spontaneous functional connectivity remains evident in intrinsic spinal networks during unconsciousness, enabling persistent communication between functionally and spatially diverse regions of the spinal gray matter

Next, we asked whether pairs of spontaneously active units exhibited correlated discharge patterns. Statistical matrices of unit-pair correlations for a 5 min epoch with a high degree of connectivity can be seen in *Figure 3*. In *Figure 3a*, each pixel's color represents the magnitude of correlation between the two units defined by an x–y pair; connection polarity is not indicated (although see *Figure 4c*). *Figure 3b* indicates the p-values of the correlations. Across all animals and epochs in the urethane cohort, 4.2 ± 0.8% of unit pairs exhibited significantly correlated temporal discharge patterns.

We then sought to determine the gross anatomical organization of synchronous unit pairs. To do so, we constructed functional connectivity maps that enabled topological aspects of the correlation structure to be visualized in the context of the microelectrode array geometry and location within the spinal cord. Because it is not possible to know if the units were synaptically coupled, we adopt the term *functional* connectivity to refer to significant temporal synchrony between unit pairs.

*Figure 4* depicts examples of such intraspinal functional connectivity maps from two representative animals. *Figure 4a, b* depicts *all* significant connections, regardless of polarity; *Figure 4c* highlights the topology of excitatory and inhibitory connections from *Figure 4a*. In *Figure 4c* (red), we show only the significant excitatory connections from the animal in *Figure 4a*; in *Figure 4c* (blue), we show putative inhibitory connections, also from the animal in *Figure 4a*. In both figures, gray circles represent each electrode on the microelectrode array, referred to as 'nodes.' Green highlighted circles in *Figure 4* were determined to be the most connected nodes of the array (see Materials and methods). Qualitatively, it is evident from *Figure 4* that pairs of temporally correlated, spontaneously active units can be found (**a**) at all sampled dorsoventral depths, (**b**) within each gross anatomical region, and (**c**) between all anatomical regions.

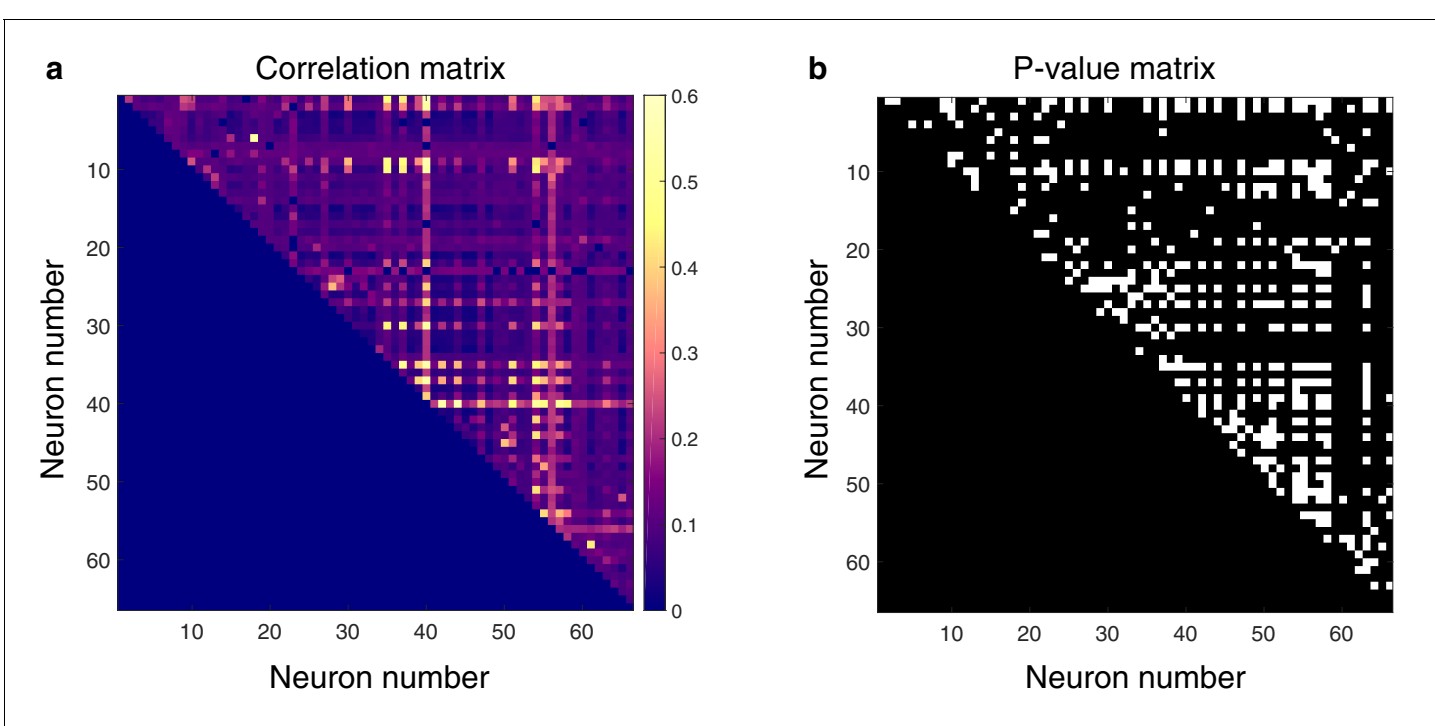

**Figure 3.** Spontaneously active units exhibit temporal synchrony. For both plots, rows and columns are ordered from 1 *N*, where *N* is the total number of units discriminated for a given trial. (**a**) Strength of temporal correlation between pairs co-active units, indicated by pixel color. Pixels below identity line are omitted because reciprocal connections were not considered. (**b**) Statistical matrix of correlation strength show in panel (**a**). White pixels represent statistically significant correlations, here defined as those with p-values ≤ 0.02. Of the 66 total spontaneously active units discriminated in this epoch, and thus 2145 possible unique connections (ignoring reciprocal connections), 438 pairs exhibited significantly correlated temporal discharge.

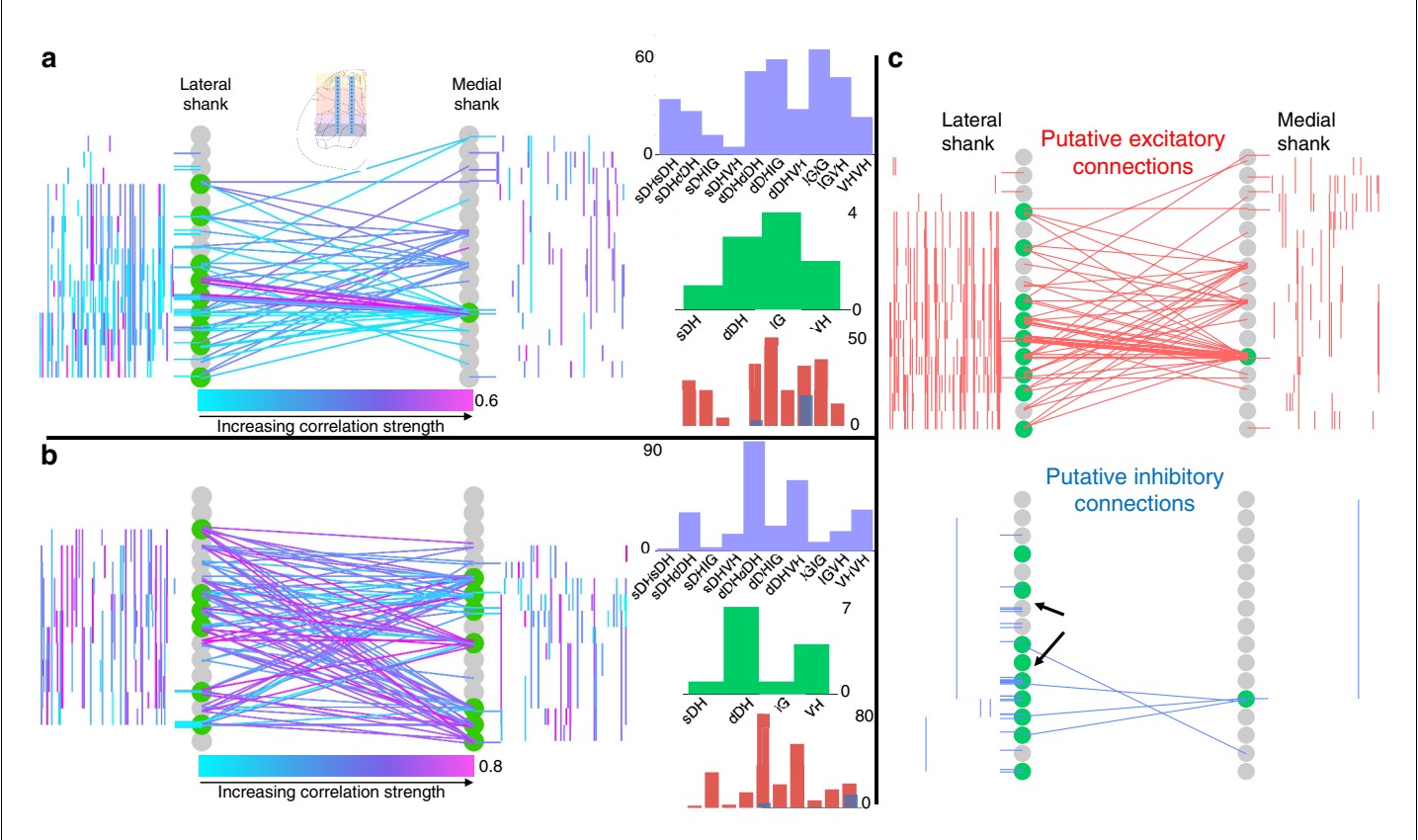

**Figure 4.** Topology of spontaneously synchronous unit pairs is not relegated to regions of primary afferent terminations, rather it links sensory- and motor-dominant regions of the spinal cord. Representative functional connectivity maps from two animals (panels **a** and **b** from same animal; panel **c** from separate animal). For all topology plots (**a–c**): spinal cord inset image in panel (**a**) shows electrode location. Gray circles represent individual electrodes on the microelectrode array. Green highlighted circles were determined to be the most connected nodes of the recording. Colored lines represent significantly correlated temporal discharge between pairs of spontaneously active units at the indicated locations (note: horizontal lines indicate connections between units discriminated from a single electrode, vertical lines are connections between units on the same shank). For panels (**a**) and (**b**), line color delineates increasing correlation strength from blue to violet; for panel (**c**), red lines indicate putative excitatory connections, blue lines indicate putative inhibitory connections. In panels (**a**) and (**b**), histograms depict the following (top to bottom): purple histograms indicate the overall anatomical distribution of significant connections (in order left to right: sDH-sDH, sDH-dDH, sDH-IG, sDH-VH, dDH-dDH, dDH-IG, dDH-VH, IG-IG, IG-VH, VH-VH); green histograms indicate the gross anatomical distribution of most connected nodes (in order left to right: sDH, dDH, IG, VH); and red/blue histograms indicate the distribution of putative excitatory and inhibitory connections, respectively, in same order as purple histograms above. Black arrows in panel (**c**), inhibitory connections, are intended simply to highlight the preponderance of within-electrode connections. sDH: superficial dorsal horn; dDH: deep dorsal horn; IG: intermediate gray matter; VH: ventral horn.

Summary functional connectivity data from all animals in the urethane cohort can be seen in *Figure 5* and *Figure 5—figure supplement 1*. The proportion of significant connections *within* regions, at 68.9%, was significantly greater than the proportion of between-region connections, 31.1% (p<0.0001; *Figure 5a*). We also found a main effect of anatomical region on the proportion of significant connections detected across all regions (F = 9.277, p<0.0001; *Figure 5a*, *Supplementary file 1*, tab 1b; *Figure 5—figure supplement 1*). This effect was driven (a) by pairs of units *within* the dDH, IG, and VH, which accounted for the highest overall proportion of connections (24.9 ± 3.6, 17.3 ± 3.7, and 17.4 ± 3.7%, respectively), and (b) by sDH-IG and sDH-VH pairs, which exhibited the lowest proportion of significant connections (1.5% and 1.2%, respectively). Predictably, the proportion of significant connections was inversely related to connection distance. For example, sDH-sDH, sDH-dDH, sDH-IG, and sDH-VH connections account for 9.3, 6.3, 1.5, and 1.2% of overall significant connections.

The gross anatomical connectivity results were also reflected in the distribution of the most connected nodes. Nodes in the dDH were classified as belonging to the most connected group in a

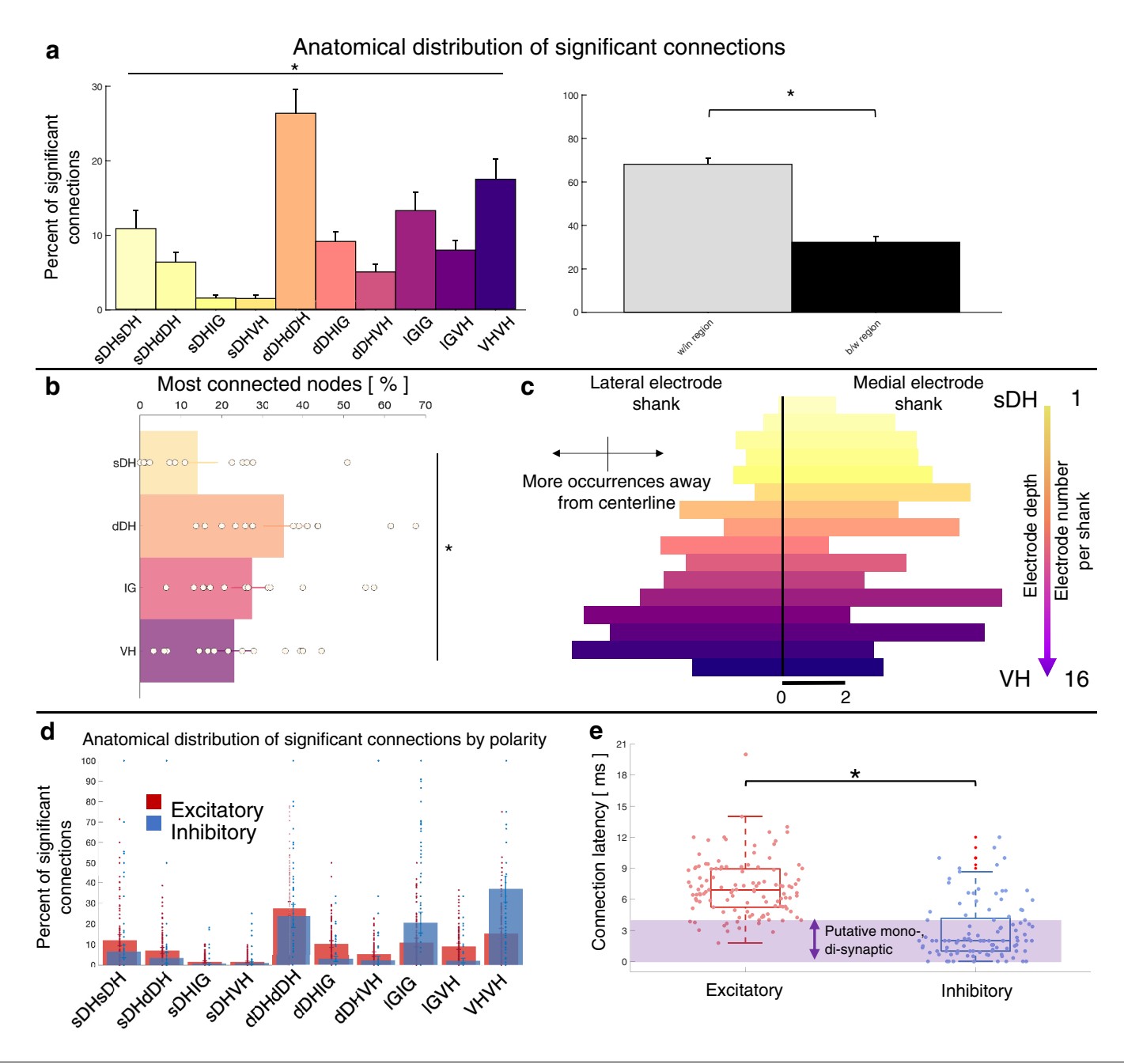

**Figure 5.** Summary of topological data for urethane-anesthetized animals. (**a**) Proportion of significant connections by anatomical region ($N$ = 13 animals). From left to right, bar plots indicate connections from sDH-sDH, sDH-dDH, sDH-IG, sDH-VH, dDH-dDH, dDH-IG, dDH-VH, IG-IG, IG-VH, and VH-VH. Darkening color gradient from left to right qualitatively indicates depth from dorsal surface of spinal cord. Grayscale plots are the proportion of within- and between-region connections, respectively. Significant connections are not uniformly distributed anatomically, with an overall main effect of connection location (p<0.0001) and significantly more within-region than between-region connections (p<0.0001). (**b**) Gross anatomical distribution of the most connected nodes ($N$ = 13 animals). From top to bottom (light to dark): sDH, dDH, IG, and VH. Significant main effect of anatomical region on proportion of most connected nodes, p=0.009. (**c**) Histogram of most connected nodes across electrodes on each shank. Bars to left of vertical black line reflect lateral electrode shank, and bars to right of vertical black line reflect medial electrode shank; from top to bottom (light to dark), each row represents one electrode (16 total rows). Bar length indicates the number of occurrences that electrode was determined to be in the 'most connected' subset. (**d**) Spatial distribution: proportion of significant connections by polarity (excitatory, inhibitory) and anatomical region. Red bars: putative excitatory connections; blue bars: putative inhibitory connections. (**e**) Temporal distribution: latencies of significant excitatory (red) and inhibitory (blue) connections. Purple shaded region intended to highlight latencies compatible with potential monosynaptic or disynaptic connections. Inhibitory

*Figure 5 continued on next page*

*Figure 5 continued*

latencies were significantly shorter than excitatory latencies on average (p=0.0003). sDH: superficial dorsal horn; dDH: deep dorsal horn; IG: intermediate gray matter; VH: ventral horn.

The online version of this article includes the following figure supplement(s) for figure 5:

**Figure supplement 1.** Summary of topological data for urethane-anesthetized animals.

greater proportion of trials (35.4 ± 4.6%) than nodes in the sDH (14.1 ± 4.3%), IG (27.4 ± 4.4%), or VH (23.0 ± 3.9%), driving an overall main effect of anatomical region on the distribution of most connected nodes (F = 4.333, p=0.009; *Figure 5b*, *Supplementary file 1*, tab 1c). It should be noted, however, that the dDH comprised a relatively larger dorsoventral extent than did the other regions, and thus contained a greater number of nodes. This contributed to the greater proportion of connections attributed to it. To this point, in *Figure 5c*, we show a histogram of the most connected nodes across the 32-channel microelectrode array. While a clear increase in counts is evident moving from dorsal-most to ventral-most, many individual electrodes in the IG or VH exhibited a higher occurrence of being 'most connected' than those in the dDH (see Discussion).

Finally, we characterized the distribution of putative excitatory and inhibitory connections. In *Figure 5d*, we highlight their anatomical distribution. We found that connections within the dDH, within the IG, and within the VH contained the highest proportion of putative inhibitory connections (22.7 ± 5.3%, 24.1 ± 7.3%, and 37.8 ± 9.0%, respectively), with the dDH containing the highest proportion of excitatory connections (25.9 ± 3.7%). Interestingly, only the dDH displayed an approximately balanced proportion of excitation and inhibition – that is, nearly the same proportion of the overall number of putative excitatory connections as overall putative inhibitory connections.

Although it is striking that the highest percentage of inhibitory connections were all within specific regions rather than between regions, this may be a practical consequence of the extracellular recording technique: detection of inhibitory connections via correlation-based approaches is notoriously challenging, in part because both cells must have a relatively high and stable base firing rate to detect a reduction in firing. Functional connectivity, which includes many polysynaptic pathways, makes detection more difficult still. Thus, some of the differences we observed in the within-vs. between-region distribution of inhibitory connections may reflect these experimental elements and should not be interpreted exclusively as a physiological feature of spinal network structure. The relative balance of inhibitory connections may also change with sensorimotor reflex activation, volitional movement, nociceptive transmission, etc., even using extracellular recording techniques.

The distribution of latencies between each statistically significant connection is shown in *Figure 5e*. Mean excitatory latency was significantly longer than the mean inhibitory latency, at 6.4 ± 0.6 ms vs. 2.7 ± 0.4 ms (p=0.0003), with both categories including latencies consistent with putative mono-, di-, and polysynaptic pathways. Interestingly, we find a subset of both excitatory and inhibitory connections with latencies between 0 and 1 ms. While some of these connections could indeed be monosynaptic and the lower-than-expected delay merely related to binning spikes, the most likely interpretation for coincidentally firing unit pairs would be a shared presynaptic input. While the distribution of inhibitory latencies contained was skewed towards an increased probability of observing putative mono- and disynaptic connections, this apparent disparity may also be related to the aforementioned challenging of detecting inhibition via extracellular recording techniques.

## Functional connectivity within and between deep regions of the spinal gray matter is not abolished by preferential pharmacological depression

The finding of robust functional connectivity between sensory-dominant dorsal horn regions and the IG and VH was unexpected. Especially intriguing was the presence of vigorous neural transmission within the IG and VH themselves. Although urethane profoundly depresses spontaneous discharge in the dorsal root ganglia, it exerts less of a depressive effect on cells deep in the gray matter (i.e., the IG and VH) (*Daló and Hackman, 2013*; *Hara and Harris, 2002*). To control for the potential influence of this anesthetic gradient on our findings, we conducted an additional set of experiments in a cohort of eight rats anesthetized with isoflurane. Isoflurane is a more potent depressant of spinal motor activity than urethane, with an overall gradient of depression that increases from the dorsal

horn to the VH (*Kim et al., 2007*). For example, while nociceptive pathways in the sDH remain largely uninhibited by isoflurane, premotor interneurons and motoneurons in the VH are markedly depressed (*Grasshoff and Antkowiak, 2006*). Mean intraspinal resting motor threshold confirmed the greater depression of VH cells by isoflurane than urethane (isoflurane threshold: 20.4 µA; urethane threshold 14.0 µA).

In total, we recorded from 484 well-isolated units across the nine rats, translating to ~51 ± 2 units per trial. The mean number of units recorded per trial did not differ between the urethane and the isoflurane cohorts (p=0.0718). A representative raster plot of spontaneous neural activity from one trial is shown in *Figure 6a*. Surprisingly, spontaneously active units were observed throughout the dorsoventral extent of the sampled region in all animals, including the IG and VH. The mean numbers of units per region are as follows: sDH: 9 ± 2, dDH: 20 ± 3, IG: 12 ± 1, VH: 13 ± 1 (main effect of region: F = 6.650, p=0.001; *Figure 6b*, *Supplementary file 1*, tab 1d). In *Figure 6c*, we show a representative functional connectivity map for the isoflurane cohort.

Summary data from the isoflurane cohort can be seen in *Figure 7* and *Figure 7—figure supplement 1*. Here, we show the gross anatomical distribution of significant connections. Similar to the urethane cohort, we observed a significantly greater proportion of connections *within* regions (66.4%) than across regions (33.6%) (p=0.005), and an overall main effect of anatomical region (e.g., sDH-sDH, sDH-dDH) on the proportion of significant connections (F = 6.517, p<0.0001; *Supplementary file 1*, tab 1e). Interestingly, despite the different mechanisms of action and

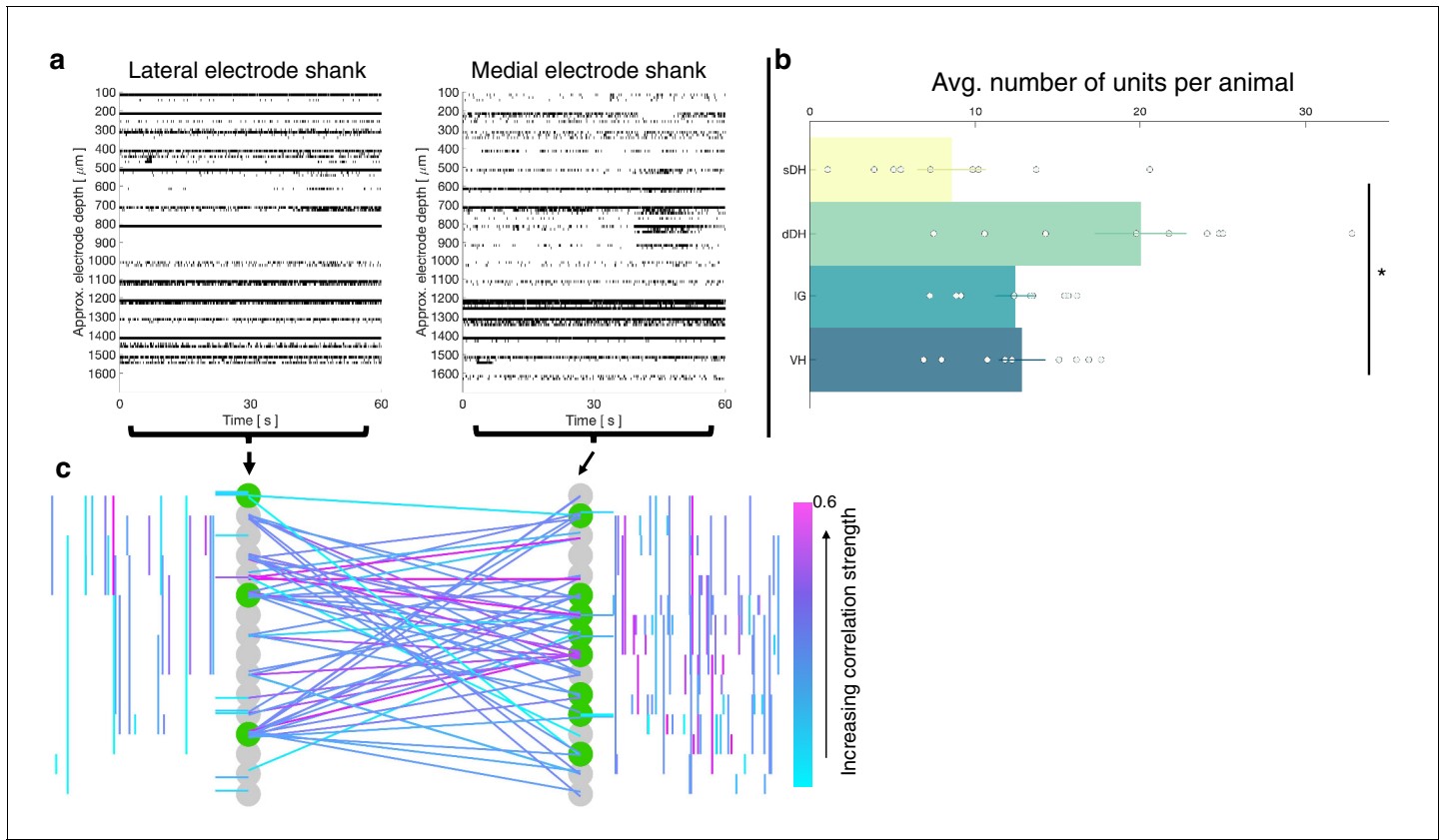

**Figure 6.** Vigorous spontaneous sensorimotor functional connectivity persists despite preferential depression of ventral horn (VH) cells. (a) Raster plot of spontaneously active neurons from a representative isoflurane-anesthetized animal. Each row of hatches represents a discrete neuron. X-axes (time) are synchronized across the two subplots. (b) Distribution of spontaneously active units per gross anatomical region across animals in the isoflurane cohort (*N* = 9 animals). The deep dorsal horn contained significantly more spontaneously active units on average than the superficial dorsal horn or VH, driving an overall main effect of region (p=0.001). (c) Representative functional connectivity map from panel (a). Gray circles represent individual electrodes on the microelectrode array (as in *Figure 4*). Green highlighted circles were determined to be the most connected nodes of the recording. Colored lines represent significantly correlated temporal discharge between pairs of spontaneously active units at the indicated locations (note: horizontal lines indicate connections between units discriminated from a single electrode, vertical lines are connections between units on the same shank). Line color delineates increasing correlation strength from blue to violet.

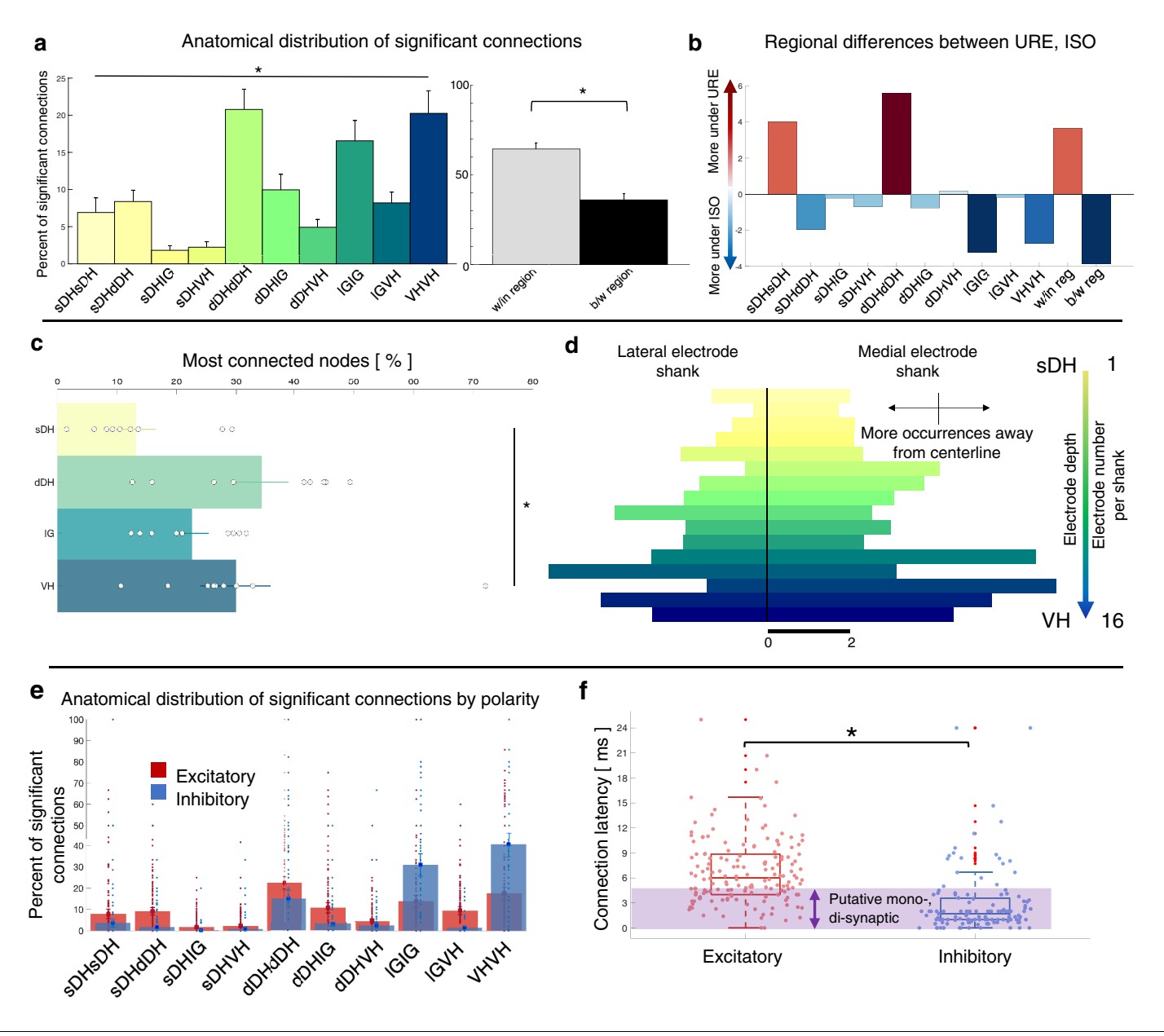

**Figure 7.** Summary of topological data for isoflurane-anesthetized animals. (**a**) Proportion of significant connections by anatomical region (*N* = 9 animals). From left to right, bar plots indicate connections from sDH-sDH, sDH-dDH, sDH-IG, sDH-VH, dDH-dDH, dDH-IG, dDH-VH, IG-IG, IG-VH, and VH-VH. Darkening color gradient from left to right qualitatively indicates depth from dorsal surface of spinal cord. Grayscale plots are the proportion of within- and between-region connections, respectively. Significant connections are not uniformly distributed anatomically, with an overall main effect of connection location (p<0.0001) and significantly more within-region than between-region connections (p<0.005). (**b**) Difference in proportion of significant connections per anatomical region between the urethane (URE) and isoflurane (ISO) cohorts. Vertical axis represents the difference in proportion of connections between the two cohorts; positive values: more significant connections in the urethane cohort; negative values: more significant connections in the isoflurane cohort. Overall, there was no statistically significant difference between the anatomical distribution of significant connections between the two cohorts. (**c**) Gross anatomical distribution of the most connected nodes (*N* = 9 animals). From top to bottom (light to dark): sDH, dDH, IG, and VH. Significant main effect of anatomical region on proportion of most connected nodes, p=0.006. (**d**) Histogram of most connected nodes across electrodes on each shank. Bars to left of vertical black line reflect lateral electrode shank, and bars to right of vertical black line reflect medial electrode shank; from top to bottom (light to dark), each row represents one electrode (16 total rows). Bar length indicates the number of occurrences that electrode was determined to be in the 'most connected' subset. (**e**) Spatial distribution: proportion of significant connections by polarity (excitatory, inhibitory) and anatomical region in the isoflurane cohort. Red bars: putative excitatory connections; blue bars: putative inhibitory connections. (**f**) Temporal distribution: latencies of significant excitatory (red) and inhibitory (blue) connections in the isoflurane

*Figure 7 continued on next page*

*Figure 7 continued*

cohort. Purple shaded region intended to highlight latencies compatible with potential monosynaptic or disynaptic connections. Inhibitory latencies were significantly shorter than excitatory latencies on average within the isoflurane cohort (p=0.017). We found no systematic differences in the spatiotemporal profiles of excitatory and inhibitory connections between the urethane and isoflurane cohorts, which preferentially depress the dorsal horns and VH, respectively. sDH: superficial dorsal horn; dDH: deep dorsal horn; IG: intermediate gray matter; VH: ventral horn.

The online version of this article includes the following figure supplement(s) for figure 7:

**Figure supplement 1.** Summary of topological data for isoflurane-anesthetized animals.

depressive profiles of the two anesthetics, we found no systematic difference in the proportion of significant connections per region across the urethane and isoflurane cohorts (anesthetic by region interaction: F = 0.369, p=0.949; main effect of anesthetic: F = 0.631, p=0.436); rather, all were within 1.8% of one another on average (range, 4–6%, *Figure 7b*). The distribution of most connected nodes in the isoflurane cohort also mirrored that of the urethane cohort. Specifically, the largest proportion of most connected nodes was found in the dDH (34.2%), the lowest in the sDH (13.2%), with 22.6% in the IG and 30.0% in the VH. There was a significant main effect of region on most connected node (F = 4.935, p=0.006; *Supplementary file 1*, tab 1f; *Figure 7c, d*). Together, these findings provide additional confirmation of the presence of persistent, synchronous discharge between functionally and spatially different regions of the spinal gray matter during unconsciousness. That such activity persisted in the IG and VH with isoflurane also underscores the apparent robustness of the finding.

The anatomical distribution of excitatory and inhibitory links also remained remarkably stable between urethane and isoflurane (*Figure 7e*). There was no main effect of anesthetic agent nor an interaction of drug by region for either the proportion of excitatory or inhibitory links in each region (excitatory: region: F = 13.981, p=0.000; region*drug: F = 0.348, p=0.819; drug: F = 0.030, p=0.865, *Supplementary file 1*, tab 1g; inhibitory: region: F = 19.403; p=0.000; region*drug: F = 0.231, p=0.794; drug: F = 0.611, p=0.444, *Supplementary file 1*, tab 1h). The mean latency of excitatory and inhibitory connections also did not change from the urethane to the isoflurane cohorts (excitatory: 6.4 ± 0.5 vs. 6.7 ± 1 ms, p=0.8188; inhibitory: 2.6 ± 0.4 vs. 3.1 ± 0.6 ms, p=0.5389). Within the isoflurane cohort, inhibitory latencies were significantly shorter than excitatory latencies (p=0.017; *Figure 7f*), which was also reflected when pooling data across both cohorts (i.e., inhibitory latencies were significantly shorter than excitatory latencies on average at 2.9 vs. 6.5 ms, p<0.0001).

## The magnitude and spatiotemporal profile of unconscious intraspinal functional connectivity is not explained by random network activity

Because these experiments characterize spontaneous rather than evoked network activity, it is reasonable to question whether the activity is likely to emerge merely by chance. To address this question, we first asked whether the proportion of significantly correlated unit pairs was greater than that which would be expected by an interconnected population of statistically matched neurons firing randomly. We addressed this question in a twofold manner, first by comparing the observed data to a synthetic dataset designed to jitter short-latency interactions and second by comparing the observed data to a synthetic dataset designed to jitter long-latency interactions. Across animals, we find that the mean proportion of significantly correlated unit pairs in the synthetic datasets was significantly lower than that observed experimentally (urethane: short-latency synthetic: 2.7 ± 0.4%, long-latency synthetic: 2.6 ± 0.5%, experimentally observed: 4.2 ± 0.8%, p=0.0053; isoflurane: short-latency synthetic: 2.7 ± 1.1%, long-latency synthetic: 2.6 ± 0.5%, experimentally observed, 3.9 ± 1.3, p=0.0033). On a per-animal level, we find that the proportion of significant connections in the observed data always exceeded its synthetic counterpart; that is, in no animals did we detect only as few (or fewer) significant connections than would be expected at random when controlling for the uniqueness of each animal's own data. These findings indicate that the overall degree of temporal synchrony was highly unlikely to be observed at random.

Next, we asked whether the spatial patterns of connectivity – that is, the topology of the significantly correlated unit pairs – differed from a random structure. Given the consistent surgical placement of our microelectrode arrays in each experiment, their known geometry, and our definitions of the approximate boundaries between gross anatomical regions in the spinal gray matter, it is

possible to directly compute the probabilities that significant connections will exist within or between regions if neurons are distributed at random. In our paradigm, the number of electrodes per region is: sDH: 8; dDH: 12; IG: 6; VH: 6. These values can be seen in *Figure 1a*. We are interested in all within- and between-region connections as a matter of combinations, not permutations (i.e., sDH-DH connections are the same as dDH-sDH connections).

Assuming that neurons are randomly (albeit uniformly) distributed throughout the sampled gray matter, the expected value for each regional comparison is the ratio of the number of electrodes represented in a given comparison to the total number of electrodes represented across all comparisons. For example, the expected percentage of dDH-dDH connections is arrived at by dividing 12, the number of electrodes in the regional comparison, by 128, the total number of electrodes represented across all combinations. Note that the overall total (128) is not the same as the number of electrodes on the array itself (32). This difference is because we are including comparisons between regions in addition to comparisons within regions. Expected values for *overall* within-region and between-region connectivity are the sum of the individual regional percentages. Theoretical probabilities are: sDH-sDH: 6.3%; sDH-dDH: 15.6%; sDH-IG: 10.9%; sDH-VH: 10.9%; dDH-dDH: 9.4%; dDH-IG: 14.1%; dDH-VH: 14.1%; IG-IG:4.7%; IG-VH: 9.4%; and VH-VH: 4.7%. For within- and between-region connections, the probabilities are 25 and 75%, respectively. We then verified that the bootstrapped synthetic data indeed converged to these theoretical predictions (*Figure 8a*).

We found an overall main effect of anatomical region on connectivity patterns between the bootstrapped synthetic data and the observed data (urethane: F = 10.571, p<0.0001, *Figure 8b*, *Supplementary file 1*, tab 1i; isoflurane: F = 7.251, p=0.001, *Supplementary file 1*, tab 1j) and, notably, a significant interaction of region by cohort (i.e., real or synthetic urethane data; F = 16.168; p<0.0001 *Supplementary file 1*, tab 1i; isoflurane: F = 11.561, p<0.0001, *Supplementary file 1*, tab 1j). Post-hoc testing across regions revealed a lower proportion of significant sDH-dDH, sDH-IG, sDH-VH, dDH-IG, and dDH-VH connections in the real compared to the synthetic dataset and a significantly greater proportion of dDH-dDH, IG-IG, and VH-VH connections in the observed compared to the synthetic dataset (*Figure 8b*). Overall, we found a significantly greater proportion of within-region connections in the observed dataset than the synthetic dataset (68.9 vs. 26.3%, p<0.0001) and a significantly lower proportion of between-region connections in the observed dataset compared to the synthetic dataset (31.1 vs. 73.7%, p<0.0001).

## Absence of latent sensory afferent feedback suggests an intrinsic spinal source of the persistent network activity

Latent sensory afferent feedback could provide an input to spinal networks during unconsciousness. If sufficiently vigorous, this feedback could lead to activation of a diverse population of spinal interneurons and confound the interpretation of whether an intrinsic spinal source gave rise to the persistent network activity we observed. To determine whether this was likely to be the case, we conducted a further set of validation experiments in two rats (*Figure 9*).

In these rats, we implanted a recording electrode around the ipsilateral sciatic nerve proximal to its bifurcation (*Figure 9a*) and recorded spontaneous baseline ENG (*Figure 9b*; as during the intraspinal recording sessions) and ENG during periods of induced cutaneous and proprioceptive sensory transmission (*Figure 9c–e*). In the anesthetized, *unblocked* state, action potential discharge was not evident during periods without sensory stimulation (*Figure 9b*). Overall ENG amplitude was also negligible during these periods, and its standard deviation remained constant.

We then mechanically probed the L4–L6 dermatomes to quantify potential differences in afferent transmission between periods with sensory stimulation and those without. Representative epochs of each type are shown in *Figure 9c, d*. In *Figure 9c*, we show ENG in response to light touch of the L4–L6 dermatomes, including over regions of glabrous and of hairy skin. Dots above the ENG are rasters of individual spikes discriminated from the compound action potential/multi-unit ENG activity. In *Figure 9d*, we show ENG during periods of induced proprioceptive transmission as we plantarflex and dorsiflex (and hold, as indicated) the ankle. In both panels, the horizontal dashed line indicates the average ENG amplitude during bursts of induced sensory transmission, and the solid horizontal line below it indicates the average ENG amplitude during periods *without* sensory stimulation plus 3× its standard deviation. During epochs between delivery of sensory stimuli, the hindlimb rested gently on a pad with the plantar surface of the hindpaw facing upwards, as during our intraspinal recording sessions (which is also the same as in *Figure 9b*). Note that no persistent or

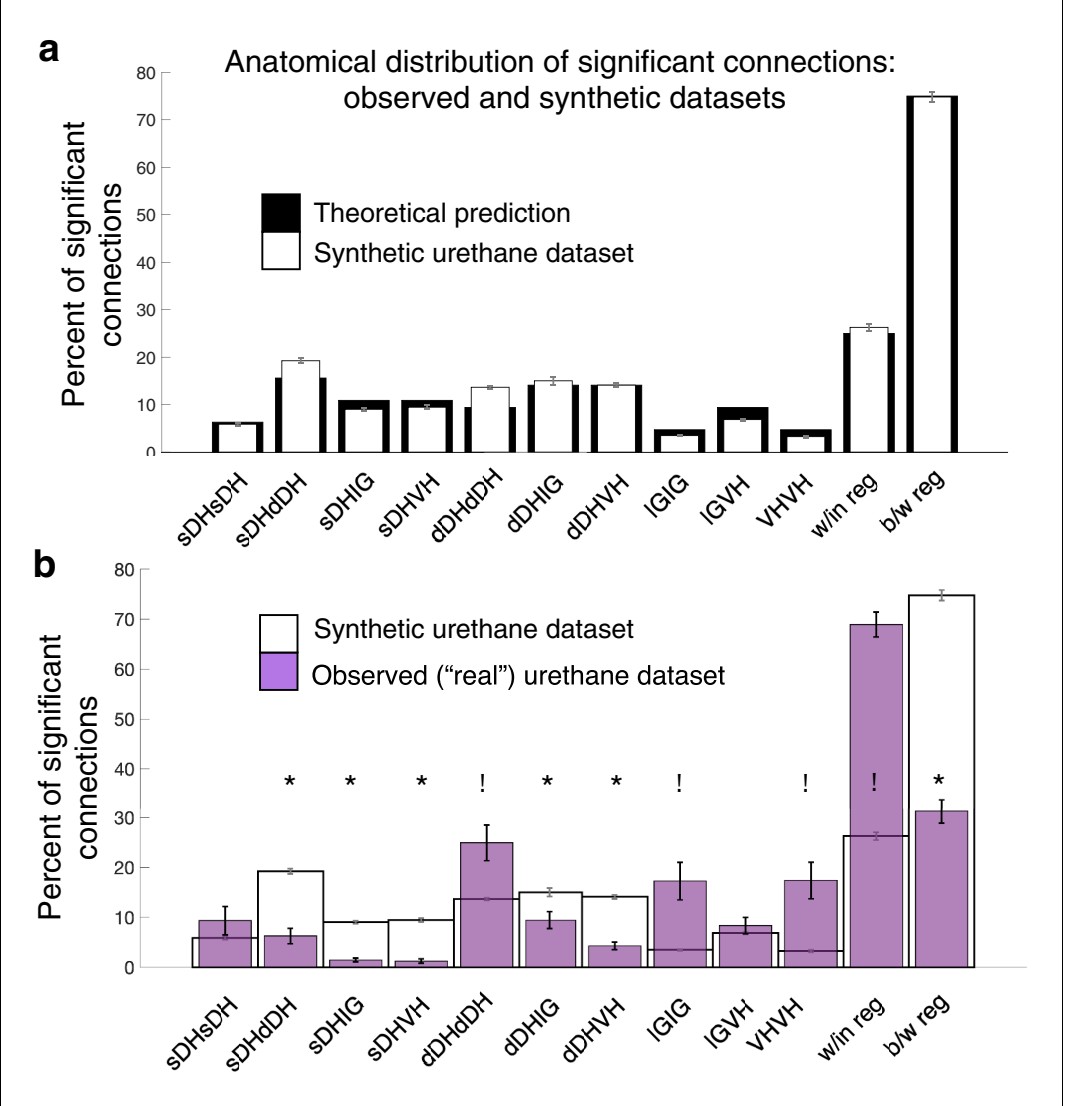

**Figure 8.** Experimentally realized spatial patterns of functional connectivity diverge from predictions of random network interactions. (**a**) Proportion of significant connections by anatomical region. From left to right, bar plots indicate connections from sDH-sDH, sDH-dDH, sDH-IG, sDH-VH, dDH-dDH, dDH-IG, dDH-VH, IG-IG, IG-VH, and VH-VH. Black bars indicate theoretical predictions; white bars indicate results of simulations ± sem (i.e., synthetic data). The synthetic dataset, generated from randomly shuffling by ±0–5 ms each spike time of each neuron in each trial, then repeating >1000 ×, converges to theoretical predictions. Theoretical predictions are based upon the number and anatomical distribution of electrodes throughout the gray matter. (**b**) Anatomical distribution of synthetic data (white, as in panel **a**) compared to experimentally realized urethane data (*N = 13*, purple bars). We found a significant interaction of cohort by anatomical region (real vs. synthetic, p<0.0001), indicating the divergence of the real dataset from that which would be expected by a population of interconnected neurons that are statistically similar but spiking at random. Asterisks indicate connections in which the synthetic data was overrepresented relative to the real data; crosses indicate connections in which the real data was overrepresented relative to the synthetic data. Most notably, we found significantly more within-region connections in the real data compared to the synthetic (p<0.0001), and significantly fewer between region connections in the real compared to the synthetic data (p<0.0001). sDH: superficial dorsal horn; dDH: deep dorsal horn; IG: intermediate gray matter; VH: ventral horn.

The online version of this article includes the following source data for figure 8:

**Source data 1.** This file contains source data for *Figure 8*.

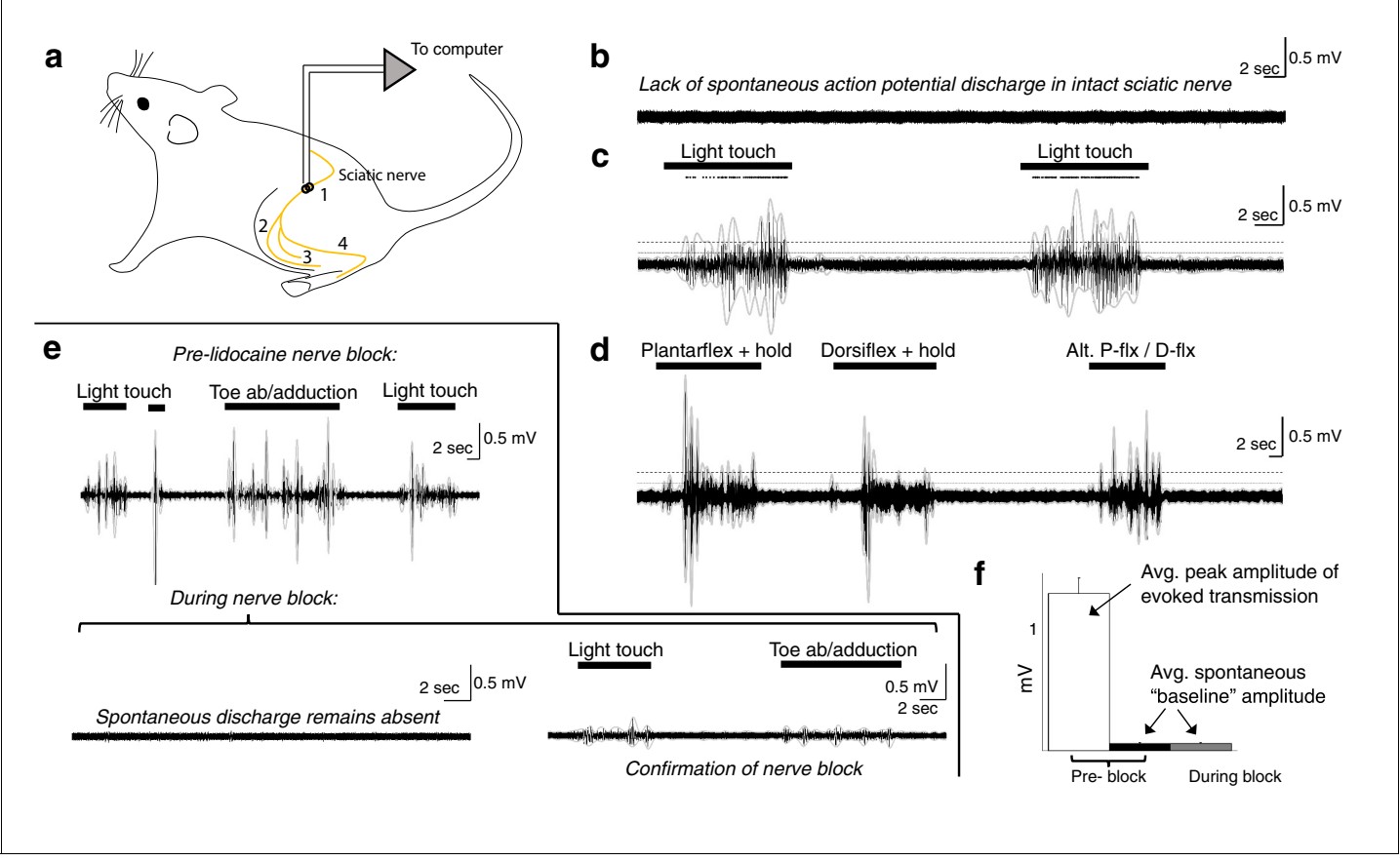

**Figure 9.** Spontaneous baseline electroneurographic (ENG) activity in the sciatic nerve is minimal and unaltered by nerve block. (**a**) Schematic diagram of recording site and relevant anatomical features. Yellow line indicates the nerve; we recorded sciatic nerve ENG using a hook electrode located at site #1, proximal to the bifurcation into tibial and peroneal nerves; site #2 represents the common peroneal nerve, site #3 represents the sural nerve, and site #4 the tibial nerve. (**b**) Representative ENG activity in the absence of sensory stimulation, as during intraspinal recording sessions. No spontaneous action potential discharge is present, and ENG amplitude is minimal and constant. (**c**) Large bursts of high-amplitude ENG are induced by cutaneous stimulation of the L4–L6 dermatomes. Stimulation epochs are indicated by the top-most horizontal bars, and the dots above the ENG are rasters of individual spikes discriminated from the compound action potential/multi-unit ENG waveforms. The horizontal dashed line indicates the average ENG amplitude during bursts of induced sensory transmission, and the solid horizontal line below it indicates the average ENG amplitude during periods *without* sensory stimulation plus 3× its standard deviation. (**d**) Identical in layout to panel (**c**), with proprioceptive feedback rather than cutaneous. The ankle was plantarflexed and held, dorsiflexed and held, and then alternated between plantarflexion and dorsiflexion. (**e**) Top panel: sciatic nerve ENG recording during periods of induced sensory transmission (horizontal black bars) and baseline transmission prior to lidocaine nerve block. Bottom panel (left): baseline ENG 30 min after epineurial lidocaine injection, showing a lack of spontaneous action potential discharge and minimal amplitude. Bottom panel (right): minimal ENG during attempted induction of sensory transmission confirms efficacy of nerve block. (**f**) Spontaneous baseline ENG amplitude is indistinguishable before (black) and during (gray) nerve block, and is 16.7× smaller than average peak ENG amplitude during bursts of induced sensory transmission before the block (white). Note: the y-axis scales are the same for all plots in (**b**–**e**).

spontaneously arising action potential discharge is evident during epochs between sensory stimulation; that is, the nerve returns to quiescence.

We then enveloped the multiunit ENG (gray lines over the ENG; 250 ms envelope window) and extracted amplitude and variability metrics for periods of spontaneous baseline ENG and induced sensory transmission. Across modalities of induced sensory transmission, bursts of multi-unit ENG had an average peak amplitude 16.7× (±0.82 sem) greater than the average amplitude of spontaneous baseline ENG ($N$ = 51 pairs). The average ENG amplitude across each period of sensory stimulation (not the average peak amplitude, as previously reported) was 3.09× (±0.17 sem) greater than the sum of the spontaneous baseline ENG amplitude and its standard deviation.

Next, we blocked afferent transmission via epineurial lidocaine injection. In the top panel of *Figure 9e*, we show periods of induced sensory transmission and interleaved periods of spontaneous baseline ENG prior to nerve block. In the bottom-left panel of *Figure 9e*, we show a representative

epoch of neural data recorded 30 min after lidocaine injection. No sensory stimulation was delivered during this epoch, and spontaneous action potential discharge expectedly remained absent. In the bottom-right panel of *Figure 9e*, we confirm the blocking effect of lidocaine injection by attempting to induce sensory transmission. Qualitatively, it is evident that the spontaneous baseline ENG is indistinguishable pre- and post-block (*Figure 9b* vs. *Figure 9e*). This similarity is represented quantitatively in *Figure 9f*, which shows that the average spontaneous baseline ENG before the block (black bar) is indistinguishable from that during the block (gray bar). *Figure 9f* also depicts the average peak ENG burst amplitude across trials (white bar) to provide reference for the observed spontaneous baseline ENG amplitude.

## Discussion

### Presence of an intrinsic spinal network active during unconsciousness

Our primary finding is that neural transmission persists in the spinal cord during unconsciousness at a level and with a structure that appears to be non-random. We interpret our findings as supporting the emerging view that the spinal cord possesses intrinsic networks that maintain functionality during unconsciousness and in the absence of evoked neural transmission (*Barry et al., 2014*; *Eippert and Tracey, 2014*).

In intrinsic *supra*spinal networks, functional neural transmission during unconsciousness involves patterned activity within local and regional circuits as well as communication between functionally and spatially distributed neural structures (*Demertzi et al., 2019*; *Fox et al., 2005*; *Greicius et al., 2003*; *Mashour and Hudetz, 2018*; *Raichle et al., 2001*; *Steriade et al., 1993*; *Wenzel et al., 2019*). Thus, we reasoned that persistence of correlated discharge at multiple spatial scales would also be a necessary precondition for intrinsic spinal networks to maintain functionality during unconsciousness. Central to this idea would be the presence of non-random functional connectivity within sensorimotor regions deep in the gray matter (in addition to connectivity within and between the predominantly sensory regions of the dorsal horn) as the spinal cord plays a key role in sensorimotor integration and motor output.

To this point, we found a greater proportion of connectivity within the VH than within or between any other region(s) except within the dDH, despite a lack of motor output. Connections within the IG were the third most represented (behind dDH-dDH and VH-VH). Of particular note is the proportion of VH-VH connections relative to dDH-dDH connections. While it is perhaps not surprising that the dDH exhibited the greatest interconnectivity given that it forms both local and distributed circuits and receives direct primary afferent input, it is however surprising that, when normalized for anatomical area, the dDH exhibits only ~60% as much within-region connectivity as the VH.

Previous studies have found resting state functional connectivity within the dorsal horns and the VH, respectively, but it has been an enduring question whether functional connectivity exists between the dorsal horn and other regions of the spinal gray matter during unconsciousness, particularly in the absence of evoked responses (*Barry et al., 2014*; *Eippert et al., 2017*; *Kong et al., 2014*; *Tl et al., 2019*). Remarkably, we found that >20% of all significant connections were between the sDH or dDH and the IG or VH (e.g., *Figure 5a*). To the best of our knowledge, this is the first such demonstration of single-neuron-level spontaneous functional connectivity between sensory and motor regions of the spinal gray matter during unconsciousness. From these findings, we can conclude that spontaneous synchronous discharge of spinal neurons during unconsciousness is not confined to local, sensory-dominant circuits in the dorsal horn; rather, it spans spatially and functionally distinct regions of the spinal gray matter, reflecting the integrative nature of spinal neural transmission during periods of wakeful behavior.

Determining whether the connectivity we see truly reflects the presence of orderly activity in an intrinsic spinal network during unconsciousness is a complex process, in part because of the potential role of sensory afferent inflow. On the other hand, the presence of nominal sensory inflow does not itself exclude the possibility that intrinsic activity was maintained; merely that the observed activity reflects the interaction of the two. This would be analogous to studies of resting state functional connectivity in the brain during inattentive wakefulness (e.g., the default mode network), where environmental stimuli and sensory feedback are continuously present, but lack saliency (*Raichle et al.,*

*2001*). Nevertheless, several lines of experimental controls and results support our conclusion that the observed connectivity was not due merely to sensory afferent inflow.

The most direct evidence in support of a primary role for intrinsic spinal sources as opposed to a primary contribution from sensory afferent feedback comes from the peripheral nerve (*Figure 9*). Recording from the sciatic nerve proximal to its bifurcation (*Figure 9a*, site #1), we found no evidence of spontaneous action potential discharge in fibers innervating the L4, L5, or L6 spinal segments (*Figure 9b*). The overall magnitude of spontaneous baseline ENG was also negligible, as evidenced by the 16.7× greater ENG amplitude observed during induced sensory transmission (*Figure 9c–f*). Importantly, the magnitude of ENG was also effectively constant when sensory stimulation was not being delivered, revealing no underlying multi-unit activity. The amplitude and standard deviation of the spontaneous baseline ENG in our preparation were also unchanged by pharmacological nerve block – a form of temporary deafferentation – further suggesting that latent sensory feedback was minimal (*Figure 9f*).

We next return to the finding of connectivity within and between the IG and VH. These regions would not be expected to receive meaningful direct afferent input in our preparation. The primary source of such input would be proprioceptive muscle and joint afferents, in particular the 1a, 1b, and group II fibers. While 1a afferents indeed synapse directly onto motoneurons, in our preparation muscle length was held constant. Activity in 1b and Group II afferents would likewise be negligible in our preparation, as muscles were not developing tension and were held in a neutral, unstrained position. Importantly, *Figure 9c and e* highlight the differences between spontaneous baseline ENG (i.e., as during the intraspinal recordings) and ENG during periods of proprioceptive feedback, revealing an absence of proprioceptive transmission outside of periods in which it was specifically induced.

Another argument against an exclusive role of sensory feedback driving our connectivity results and in support of a role for persistent activity in an intrinsic network is that sDH and dDH connectivity was robust in animals anesthetized with urethane. As mentioned in Results, we chose urethane specifically for its documented ability to block spontaneous dorsal root activity (*Daló and Hackman, 2013*; *Hara and Harris, 2002*). It is also worth reiterating that we chose an electrode implantation site whose corresponding dermatome primarily included the glaborous skin of the plantar surface of the hindpaw. This region had no physical contact with the surgical field, instruments, etc., further minimizing undue afferent feedback. Although physical deafferentation would have wholly eliminated natural sensory afferent activity, it could have paradoxically increased discharge in the residual dorsal roots, second-order neurons, or local dorsal horn neurons (*Eschenfelder et al., 2000*).

A counterpoint to this interpretation would be that the activity we observed within and between the IG and VH is related to polysynaptic activation of premotor interneurons and other interneurons intercalated amongst motor pools from latent connections to the sDH and dDH. We addressed this potential confound by characterizing functional connectivity in a separate cohort of rats anesthetized with isoflurane, an anesthetic known to preferentially depress VH cells relative to the dorsal horn cells, including premotor interneurons (*Kim et al., 2007*; *Kohno and Wakai, 2005*). We found that functional connectivity in the IG and VH (as well as the sDH and dDH) persisted largely unchanged in animals administered isoflurane, and therefore choice of anesthetic agent could not explain our findings. In fact, we find the spatiotemporal patterns of connectivity to be remarkably consistent across the two anesthetic agents. This finding, in conjunction with other experimental controls, further supports the notion that the results are not merely an epiphenomenon or primarily reflective of afferent transmission.

Separate from afferent feedback, some degree of spontaneous, possibly random, neural transmission would presumably be expected in the spinal cord regardless of whether a structured intrinsic network is active during unconsciousness. Therefore, it was important to understand how the observed proportion of functionally connected units and their topology compared to that which might be expected in populations of statistically similar interconnected neurons spiking randomly. We developed a series of large synthetic datasets to address these questions, finding an average of 105% *more* pairs of functionally connected units across rats in the observed compared to the synthetic datasets. This comparison is the most direct application of our synthetic datasets, and our findings indicate that the observed proportion of functionally connected units was unlikely to have occurred by chance. It also reinforces the view that the spinal cord indeed possesses intrinsic networks active during unconsciousness, which could be involved in multimodal neural processing.

Regarding topological aspects of the correlated units, we also find a marked departure from a random structure. One of the most pronounced topological features of the observed data, particularly compared to theoretical benchmarks, was the different proportion of within-region vs. between-region connectivity. Indeed, we found significantly greater within-region connectivity than between-region connectivity overall (~70 vs. ~30%), opposite our prediction. However, interpreting whether the observed between-region connectivity is more or less than what would be expected at random is challenging because of uncertainties introduced by the nature of our experiments, and we urge caution in this regard. The primary reason for this uncertainty is that it is not possible to know what percentage of the total population of spontaneously active neurons our sample represents, which may also be affected by anesthetic depth – a variable we did not systematically explore. Comprehensively estimating what types of neurons and anatomical regions are most likely to exhibit spontaneous activity is likewise not possible.

From a practical standpoint, the apparently increased prevalence of within-region connections compared to between-region connections could also be driven in part by the sDH. While the sDH contains the most theoretical between-region connections, it is a particularly challenging region to study in vivo using implanted microelectrode arrays. Indeed, its proximity to the electrode insertion site increases the likelihood of tissue damage, which is compounded by the small size and fragility of the cells it contains (e.g., in the substantia gelatinosa). The sDH also contains a preponderance of between-region circuits dedicated to transmission of nociceptive neural activity from the periphery, but nociception was not a component of our protocol. These considerations presumably reduced the overall proportion of between-region connections we observed, which was shifted further towards a majority of within-region connections by the fourfold overrepresentation of VH-VH connections.

Several other factors also contribute. For example, in any neural system, one would predict increased synchrony amongst spatially co-localized neurons and less synchrony between spatially distant neurons. Thus, our ability to detect between-region connectivity using individual spike trains is presumably not uniform. The lack of overt stimuli and neural drive in our preparation also suggests that the discharge rate of spontaneously active units was likely lower than it would have been during awake, behaving conditions, rendering temporal correlation analyses more difficult due to fewer chances to observe coincident spikes. An additional consideration is that we did not characterize or predict higher-order connectivity patterns (e.g., 3, 4, 5 link connections). Together, these factors would have contributed to a preferential *underestimation* of between-region connectivity compared to within-region connectivity and could underlie a portion of the differences we see in the observed vs. synthetic datasets.

Thus, while we can conclude that the observed proportions of regional connectivities are non-random, non-zero, and that multiple local and distant regions are functionally connected (i.e., rejecting the null hypotheses of the analyses), we cannot delineate the specific pathways through which these connections are mediated nor can we directly contextualize the relative magnitude of the proportions themselves. Despite all of the above considerations, however, it is worth reiterating that approximately one in every five observed connections spanned sensory-dominant and motor-dominant regions.

## Possible function(s) of neural transmission in intrinsic spinal networks during unconsciousness

One potential explanation for the presence of persistent activity during unconsciousness could be reactivation of salient experience-dependent patterns of neural transmission to stabilize circuit-level synaptic connectivity. During sleep, for example, specific patterns of hippocampal and cortical activation emerge that mirror those experienced during wakefulness (*Puentes-Mestril and Aton, 2017*; *Wei et al., 2016*). Persistence of these patterns is believed to be integral to memory encoding and consolidation. It is reasonable to think that such a mechanism might be a generalized feature of complex neural circuits.

Several of our findings are consistent with this idea and suggest putative mechanisms by which it could occur. First, our finding of functional connectivity between superficial and deep regions indicates that the pathways nominally required for stabilization of multimodal patterns of neutral transmission remain active during unconsciousness. Next, we find a substantial portion of connection latencies compatible with mono- and disynaptic interactions, offering a link between broad,

network-level neural synchrony and the millisecond-timescale synaptic interactions necessary for driving plasticity and shaping behavior (*Brzosko et al., 2019*; *Feldman, 2012*). And finally, we show that both excitatory and inhibitory connections with the full complement of latencies are widely distributed throughout the gray matter, providing another mechanism for bidirectional modification of synaptic interactions (besides spike-timing-dependent plasticity) to precisely shape circuit-level neural transmission and behavior.

Although our study cannot confirm or refute whether this is indeed the purpose of the persistent network activity we observed, it is a useful framework for developing new hypotheses to probe this potential functionality. For example, we would hypothesize that if a specific salient pattern of neural transmission was introduced and reinforced prior to unconsciousness, whether naturally or as part of a targeted, plasticity-promoting rehabilitation intervention (*Jo and Perez, 2020*; *McPherson et al., 2015*; *Thompson et al., 2013*), we may find evidence of this pattern in the topology of active neurons during unconsciousness. We would also hypothesize that specific patterns of functional connectivity during unconsciousness may play a role in the chronification process after trauma or disease. Here, network activity could potentially lead either to adaptive or maladaptive reinforcement of (in) appropriate patterns of neural activity, contributing to amelioration or persistence of debilitating sensory and motor impairments (e.g., spinal cord injury-related neuropathic pain; movement impairments after stroke, spinal cord injury, or multiple sclerosis).

Other possible functions of persistent spontaneous connectivity during unconsciousness also exist. For example, it could reflect latent activity in spinal central pattern generators (although evidence for unconscious activity in these circuits has yet to be introduced to the literature). Alternatively, it could play a role in mediating inattentive physiological processes, qualitatively analogous to the default mode (or task-negative) network in the brain (*Fox et al., 2005*; *Greicius et al., 2003*; *Raichle et al., 2001*) or interoceptive networks (*Damasio and Carvalho, 2013*; *Gilam et al., 2020*; *Sternson, 2020*). However, it is difficult to extrapolate our results to these latter two constructs because we interrogated rather granular connectivity within a single spinal segment and did not directly consider transmission between spinal and supraspinal centers or sympathetic outflow. Studies of spinal BOLD signaling may offer additional evidence in support of or against these theories. It is also possible that the persistent spontaneous activity is not directly involved in synaptic stabilization or in maintenance of ongoing physiological processes. Rather, it may reflect a nominal basal state of activity required simply to prevent undue extinction of learned patterns of neural transmission (*Dunsmoor et al., 2015*). Nevertheless, our results suggest that structured spontaneous activity during unconsciousness is a fundamental property of complex neural systems and is not confined to supraspinal networks.

## Acknowledgements

This work was funded by the National Institutes of Health grants 7R01-NS111234 and K12-HD073945, both to JGM.

## Additional information

### Funding

| Funder | Grant reference number | Author |
| --- | --- | --- |
| National Institute of Neurological Disorders and Stroke | 7R01NS111234-02 | Jacob Graves McPherson |
| Eunice Kennedy Shriver National Institute of Child Health and Human Development | K12HD073945 | Jacob Graves McPherson |

The funders had no role in study design, data collection and interpretation, or the decision to submit the work for publication.

## Author contributions
Jacob Graves McPherson, Conceptualization, Resources, Data curation, Software, Formal analysis, Supervision, Funding acquisition, Validation, Investigation, Visualization, Methodology, Writing - original draft, Project administration, Writing - review and editing; Maria F Bandres, Data curation, Software, Formal analysis, Validation, Investigation, Visualization

## Author ORCIDs
Jacob Graves McPherson  https://orcid.org/0000-0002-4554-7531

## Ethics
Animal experimentation: This study was performed in accordance with the guidelines of the Institutional Animal Care and Usage Committees (IACUC) of Florida International University (FIU) and Washington University in St. Louis School of Medicine (WUSM). The studies were approved under IACUC protocols: 16-049 and 19-013 at FIU and 19-1052 at WUSM. All experiments were performed under deep, surgical grade anesthesia and animals were humanely euthanized in accordance with American Veterinary Medical Association Guidelines.

## Decision letter and Author response
Decision letter https://doi.org/10.7554/eLife.66308.sa1
Author response https://doi.org/10.7554/eLife.66308.sa2

## Additional files

### Supplementary files
• Source data 1. This file contains source data for *Figures 2* and *5*.

• Source data 2. This file contains source data for *Figures 6* and *7*.

• Supplementary file 1. Detailed statistical results for data presented in the text. Enclosed are raw ANOVA (et al.) tables for all statistical comparisons presented in the article. Each table is individually labeled as indicated in the main text and located on a separate tab within the workbook.

• Transparent reporting form

### Data availability
All data analyzed for this study are included in the manuscript and supporting files, including raw data superimposed upon group data in images. Source data for Figures. 2,5,6,7, and 8, as well as detailed statistical tables, are also included as supporting files.

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
