## [Decision Letter]

**Acceptance summary:**

Spontaneous temporally correlated neural activity in the mammalian central nervous system in the absence of sensory stimulus-evoked activity is a feature of supraspinal circuits, but less clearly established as a property of spinal cord circuits in unconsciousness when there is no motor output. In this paper, the authors convincingly provide novel direct cellular-level evidence of robust and temporally correlated spontaneous neuronal activity in the in vivo lumbar spinal cord of anesthetized unconscious rats, suggesting that such activity may be a general property of circuits throughout the central nervous system.

**Decision letter after peer review:**

Thank you for submitting your article "Spontaneous neural synchrony links intrinsic spinal sensory and motor networks during unconsciousness" for consideration by *eLife*. Your article has been reviewed by 3 peer reviewers, including Jeffrey C Smith as the Reviewing Editor and Reviewer #3, and the evaluation has been overseen by Lu Chen as the Senior Editor.

The reviewers have discussed their reviews with one another, and the Reviewing Editor has drafted this to help you prepare a revised submission. Below are essential revisions that need to be addressed in a substantially revised manuscript that will be re-reviewed.

Essential Revisions:

1) In the abstract and throughout the manuscript, the word "purpose" or "purposeful" should be replaced. It has inappropriate, unjustified, and unnecessary implications and connotations. Appropriate, non-freighted replacements are terms such as "non-random," "functional," "patterned." In the Intro and Discussion, the manuscript can then consider, as it does, the possible functional effects or outcomes of this non-random activity. For similar reasons, "network policy" in line 24 should be replaced. For example, Lines 24-5 might read: "…is consistent with the hypothesis that salient, experience-dependent…"

Methods and Results

2) Spike sorting is a procedure for detecting and identifying the extracellular action potentials of individual neurons in multi-neuron recording. A variety of algorithms have been applied for this purpose. The authors used the unsupervised wavelet-based clustering method developed by Quiroga and colleagues. Like similar algorithms, it requires manual analysis by the experimenter to verify and adjust the results. The authors discriminate ~55 neurons per trial, which translates to about two neurons from each recording electrode. Because the sorting procedure requires manual intervention to exclude false positives, perhaps the authors could provide references indicating that this is a reasonable yield per electrode. It would also be worthwhile for them to provide more information on the parameters used in the sorting procedure and possibly to compare the results with those of a more standard method (e.g., PCA).

3) Figure 1 shows the experimental setup and design, using actual data and their analyses. The figure is hard to understand because it is missing the temporal units for the raw-data trace, the raster plot, and the histograms. It is not clear whether the 4 red/orange action potentials are 4 different neurons or two neurons each detected from two electrodes. Also unclear are the meanings of the red lines and black arrows with the histograms.

4) The comparison between the real and the synthetic data is a crucial aspect of the paper, and the authors should provide more information and attention both in the Methods and the Discussion sections. It is not sufficiently clear how the data were shuffled and reconstructed to provide the synthetic data. What procedures were used to ensure that the synthetic data statistically matched the observed data? The main purpose of the synthetic data was to determine whether the connections found are likely to emerge merely by chance. The author stated that it is possible to directly compute the probabilities that significant connections will exist within or between regions if neurons are distributed at random. How have these probabilities been calculated? How did the authors verify that the bootstrapped synthetic data converged to the theoretical predictions?

5) Lines 495-542: The synthetic data do yield substantially percentages of connectivities, averaging roughly two-thirds the values for the actual data. Furthermore, it is of considerable concern and puzzlement that the synthetic connections for a number of regions are greater than the actual connections (Figure 8.b). What does this mean? For example, Figure 8 appears to show that dorsal and ventral horns are less connected in the actual data than in the supposedly random synthetic data. This seems to contradict the conclusion stated in the abstract that "we…demonstrate that spontaneous functional connectivity also links sensory and motor-dominant regions during unconsciousness."

6) The authors attribute the patterns of spontaneous activity found to reflect intrinsic spinal circuit activity, while acknowledging the possibility of sensory afferent feedback contributing to the spontaneous activity despite urethane anesthesia and isoflurane anesthesia in another experimental cohort. It would be important for the authors to discuss whether they have assessed if the spontaneous activity patterns are affected by deeper anesthetic levels than with the standard dose of urethane used for these studies. Also, despite the authors' arguments about some potential disadvantages of deafferentation, this is still an effective way to determine if there are any contributions of local afferent inputs after positioning the microelectrode arrays, particularly since they have confined their electrophysiological recordings to a single lumbar spinal segment and local deafferentation could readily be implemented. Additional information in this regard would be important to strengthen the authors' arguments about the recorded activity reflecting primarily intrinsic spinal circuit activity.

7) Unless motoneurons are completely inactive under the anesthesia, might high-amplitude spikes in VH identify likely motoneurons? If that is the case, it would be very interesting to assess motoneuron connectivities with other areas and neuronal populations. This deserves discussion.

---

## [Author Response]

Essential Revisions:1) In the abstract and throughout the manuscript, the word "purpose" or "purposeful" should be replaced. It has inappropriate, unjustified, and unnecessary implications and connotations. Appropriate, non-freighted replacements are terms such as "non-random," "functional," "patterned." In the Intro and Discussion, the manuscript can then consider, as it does, the possible functional effects or outcomes of this non-random activity. For similar reasons, "network policy" in line 24 should be replaced. For example, Lines 24-5 might read: "…is consistent with the hypothesis that salient, experience-dependent…"

We thank the reviewers for the recommendations regarding word choice and specificity; such matters are indeed critical aspects of technical writing, particularly in a field as diverse and nuanced as neuroscience.

We are in agreement with the reviewers on the intended meaning of “purposeful” and the proposed alternatives. That is, our use of the term “purposeful” was in fact meant to be operationally defined by and synonymous with the alternatives articulated by the reviews – “non-random,” “functional,” and “patterned.”

Our thought process on the choice of “purposeful” was thus: we deemed it to be somewhat more specific than “non-random,” in the sense that – at least to us – it more directly implies there may be an underlying function. While “patterned” would *strictly* indicate a lack of randomness, we felt that this somewhat subtle point may be overlooked by some readers. We view “purposeful” as closest to “functional,” and our choice of the former rather than the latter is merely a reflection of our own diction and writing style; in terms of our meaning and intent, they are interchangeable.

Changes made to the manuscript: We have updated the abstract and manuscript to replace the word “purposeful” with the most appropriate of the proposed alternatives in each instance. We have also adjusted the wording of line 24 to remove “network policy.”

Methods and Results2) Spike sorting is a procedure for detecting and identifying the extracellular action potentials of individual neurons in multi-neuron recording. A variety of algorithms have been applied for this purpose. The authors used the unsupervised wavelet-based clustering method developed by Quiroga and colleagues. Like similar algorithms, it requires manual analysis by the experimenter to verify and adjust the results. The authors discriminate ~55 neurons per trial, which translates to about two neurons from each recording electrode. Because the sorting procedure requires manual intervention to exclude false positives, perhaps the authors could provide references indicating that this is a reasonable yield per electrode. It would also be worthwhile for them to provide more information on the parameters used in the sorting procedure and possibly to compare the results with those of a more standard method (e.g., PCA).

Regarding the number of spikes per electrode: The wavelet-based algorithm implemented by Quiroga and colleagues were initially developed with 3 distinct neurons (spike shapes) per “electrode” in the simulated datasets (Quiroga et al., 2004). As such, refinement of the spike feature extraction process, classification algorithms, and overall performance assumed that extracellular arrays would access >2 neurons per electrode. Across the literature, it has been estimated that ~1-4 neurons are typically reported per electrode using spike sorting algorithms (Rey et al., 2015). Using the wave_clus algorithm in particular, 5-6 neurons can be discriminated with effectively no missed clusters or false positives; performance degradations become evident at ~8 neurons per electrode (although this typically manifests as missed clusters rather than false positives), and approximately 10 neurons per electrode has become the *de facto* standard ceiling for single-electrode discrimination (Pedreira et al., 2012; Rey et al., 2015).

Regarding the comparison to PCA: We were attracted to the wavelet-based approach of Quiroga for several reasons with respect to the alternative of PCA. From a high-level standpoint, benefits of the wavelet approach are that it includes both a time and frequency-based representation of the data and it does not assume stationarity. More specific to the performance of the unsupervised wavelet-based approach vs. PCA, in their initial report of the wavelet-based approach (available as the open-source “wave_clus” package), PCA is the default comparator. Across all of the example datasets and noise levels tested, PCA resulted in an average of 4.7X *more* classification errors than wavelet-based decomposition for the clustering approach that we used in this work. The wavelet-based approach is also particularly advantageous when attempting to discern very small differences in action potential waveform shapes. In the wavelet approach, this information is parsed into several individual wavelet coefficients that are localized in time. In PCA, this information is typically contained in the first three principal components. The challenge is that the first three principal components may not be the most well-suited for clustering, and the components with lower eigenvalues – which may be the most useful for separating spike shapes into distinct clusters – are typically not included the clustering analyses.

As a final note on the performance of the wavelet-based method vs PCA, for standard supervised approaches using K-means clustering, the rate of classification errors is more comparable between wavelets and PCA, with PCA resulting in an average of 1.1X more classification errors across noise levels. However, given that we have no a priori knowledge of how many clusters to expect for a given electrode, the supervised approach is less well suited for our application. This aspect of the process is compounded by the fact that the supervised approach will assign all detected spikes to a cluster, potentially resulting in misassignments if the initial cluster number is incorrectly assumed.

Changes made to the manuscript: We have updated the methods section to include more information about the parameters we used in the spike sorting process. Specific text is copied below for convenience.

“Single-unit neural activity was discriminated offline using an unsupervised, wavelet-based clustering approach [parameters: bandpass filter: 1Hkz – 15KHz; minimum detection threshold: 4 standard deviations (SD) from mean; maximum detection threshold: 25 SD; detection thresholds on both positive and negative deviations; filter order for detection: 4; filter order for sorting: 2]. (Quiroga et al., 2004) The veracity of discriminated units was verified manually. […] Exclusion criteria were both quantitative (e.g., predominance of ISI < 2msec) and qualitative (e.g., non-physiological shape, inappropriate action potential duration).”

We have also included new citations to support the average number of neurons we are able to discriminate per electrode. These citations can be found in the Results section, and are copied below for convenience.

“First, we quantified the gross anatomical distribution of spontaneously active units. In total, we recorded from approximately 860 well-isolated units across 13 urethane-anesthetized rats, averaging 66 ± 8 units per trial (e.g., Figure 1a). […] They are also consistent with subsequent studies using wave_clus with similar electrodes to ours, which show that 1-4 units are typically discriminated and that 5-6 units can be discriminated effectively without missed clusters or false positives (Pedreira et al., 2012; Rey et al., 2015).”

3) Figure 1 shows the experimental setup and design, using actual data and their analyses. The figure is hard to understand because it is missing the temporal units for the raw-data trace, the raster plot, and the histograms. It is not clear whether the 4 red/orange action potentials are 4 different neurons or two neurons each detected from two electrodes. Also unclear are the meanings of the red lines and black arrows with the histograms.

We appreciate the careful attention to detail here, as well. The lack of time scale is a notable oversight on our part. Regarding the action potential waveforms, these are indeed four separate neurons discriminated from a single channel on the electrode array. The histograms are examples of the cross correlation between pairs of units, with the red line indicating 0 msec lags.

Changes made to the manuscript: (1) We have included time scales for all subplots; (2) we have labeled the individual neuron action potential waveforms as “Unit a,” “Unit b,” etc.; (3) we have updated the histograms to include a label of what is depicted (xcorr(a,b)); and (4) we have updated the figure legend to explicitly state these points as well as the meaning of the histograms/cross correlation plots and associated red bars.

4) The comparison between the real and the synthetic data is a crucial aspect of the paper, and the authors should provide more information and attention both in the Methods and the Discussion sections. It is not sufficiently clear how the data were shuffled and reconstructed to provide the synthetic data. What procedures were used to ensure that the synthetic data statistically matched the observed data? The main purpose of the synthetic data was to determine whether the connections found are likely to emerge merely by chance.

We are happy to provide additional details about the synthetic dataset. First, a note of clarification: our use of the term “jittering” is distinct from what some other researchers refer to as “shuffling.” As we define it, shuffling refers to an operation where the *sequence of trials* is randomly rearranged and correlation analyses are performed again (but individual neuron spike times within a given trial are left unperturbed). Jittering, by comparison, is when the spike times of each neuron in a given trial are randomly shifted, or perturbed, but the sequence of trials is maintained.

There are several reasons why we do not use the shuffling procedure on these data. First, we are performing within-trial correlation analyses specifically to look at the potential correlation of simultaneously evolving spike trains relative to one another. Second, we do not attempt to track neurons across trials, thus we cannot assume that shuffling will result in the same neuron pairs being compared. And third, we have no reason to assume that the mean firing rate of a given neuron is stationary across trials, even though we are investigating spontaneous action potential discharge during unconsciousness. All that being said, it is possible to computationally shuffle the *anatomical location* of neurons in a given trial (as opposed to their sequence), and to this point please refer to the forthcoming paragraphs on computation of within and between region connection probabilities.

Regarding our jittering procedure and development of the synthetic dataset: For reference, the process we used for developing jittered synthetic datasets (or “surrogate” datasets, as they are sometimes called) and using them for statistical inference about neural synchrony has been formalized in review from Amarasingham et al. (Amarasingham et al., 2012).

To begin, envision a single trial for a single animal. The electrode array is implanted into the spinal cord and we record neural data on 32 electrode channels for a given amount of time. After performing spike sorting on each electrode channel, we are left with *N* spike trains, where *N* corresponds to the number of neurons discriminated during the spike sorting process for that trial. We then randomly select a value of -5, -4, -3, -2, -1, 0, 1, 2, 3, 4, or 5 milliseconds from a uniform distribution and add that value to the first spike time for, say, Neuron 1. For example, if the first spike of Neuron 1 occurred exactly 1 sec after recording commenced, and we randomly drew a value of +3msec, we would restate the first spike time for Neuron 1 as occurring at 1.003 sec after recording began. We then randomly draw another number from the full [-5:5] distribution (numbers are replaced after each draw), and add that value to the 2^nd^ spike time of Neuron 1. This process continues for all spike times for Neuron 1 during the trial. We refer to this process as jittering. The same process is subsequently performed for *all* neurons discriminated in that trial, resulting in *N* jittered spike trains. After jittering all spike times for all neurons for a single trial, we arrive at a “synthetic” trial, which we will call “Trial_syn_.” We then re-run the correlation analyses on the jittered spike trains in T_syn_. By repeating this process 1,000x for each neuron, trial, and rat, we are able to develop a large synthetic dataset from which statistical confidence intervals can be derived and hypothesis testing can be performed.

Two important features follow from this jittering manipulation that speak to the statistical similarity of the real and synthetic datasets:

1. For a given Trial_syn_, each neuron retains the same number of spikes as its empirically observed counterpart. As a result, this manipulation does not introduce confounds in terms of the number of spikes used in the correlation analyses. For example, were the number of spikes to be artificially reduced, the strength of connectivity between two neurons could be underestimated simply due to the lower number of potential occurrences of synchronous firing between neurons. Alternatively, if the jittering procedure increased the number of spikes, it could in principle artificially elevate correlations.

2. The [–5:5] jittering procedure preserves each neuron’s firing rate on a broad timescale, while randomly perturbing firing rates at short timescales. As a result, this manipulation serves as a particularly stringent test of our estimates of putative mono-, di-, and minimally polysynaptic connections, which occur at short timescales. In other words, it effectively tests the null hypothesis that no pairs of neurons are synaptically connected with mono-, di-, or minimally poly-synaptic connections. We specifically biased our jitter window towards this fine timescale (rather than a long timescale more representative of highly polysynaptic pathways) for two reasons (1) the presence of short-latency connectivity is more likely to be associated with direct communication between neurons/networks, and (2) evidence of longer timescale correlations has already been introduced by Barry et al. 2014 using fMRI (i.e., the foundation of this Research Advance article), and as such represents a more confirmatory rather than novel contribution to our understanding of spinal physiology.

However, to this latter point and in response to the reviewers’ question, we have created an additional synthetic dataset and conducted further simulations preferentially targeting longer timescale interactions, ±50msec (also with 1 msec granularity, i.e., draw a random number from [-50, -49, -48, … 48, 49, 50] msec). In this simulation we find that 2.6±0.5% of connections in the synthetic data are statistically significant, indistinguishable from that of our previous analysis of short-term interactions (2.7±1.1%), and still significantly lower than the observed data (4.2±0.8% for urethane, 3.9±1.3% for isoflurane). This reaffirms the finding of increased spontaneous functional connectivity compared to expectations.

Changes made to the manuscript: We have updated the Synthetic Data section of the Methods to include the following information:

“We generated large synthetic datasets that matched the broad statistical properties of our observed data to use as an additional means of comparison and analyses (Figure 1b). […] The overall number of spikes per unit was not changed in either jittering procedure so as to avoid confounds in the interpretation of our correlation results.”

We have also included the new long-latency simulation data in the Results section of the manuscript.

The author stated that it is possible to directly compute the probabilities that significant connections will exist within or between regions if neurons are distributed at random. How have these probabilities been calculated?

We apologize that this aspect of the methods was not made clear. As we note in the manuscript, direct computation of connection probabilities is based exclusively on the number of electrodes per anatomical region. It is not based on approximate cell density in each lamina, which would be ideal but is unfortunately infeasible. The regional connectivity analyses also necessarily assume that the number of potentially accessible neurons per electrode is approximately equal on average across all regions. While this assumption may not hold from a strict neuroanatomical standpoint, the disproportionately high density of neurons in any region of the gray matter compared to our relatively low ability to discriminate individual neurons suggests that it is a reasonable assumption to make operationally.

In our paradigm, the number of electrodes per region is: sDH: 8; dDH: 12; IG: 6; VH: 6. These values can be seen in Figure 1a. We are interested in all within and between-region connections as a matter of combinations, not permutations. In other words, we take sDH → dDH connections to be the same as dDH → sDH connections. In Author response table 1, we show the specific regional connectivities in which we are interested as well as the number of electrodes represented across each region (or regions):

“Assuming that neurons are randomly (albeit uniformly) distributed throughout the sampled gray matter, the expected value for each regional connectivity becomes the ratio of the number of electrodes represented in a given comparison to the total number of electrodes represented across all comparisons. […] Expected values for *overall* within-region and between-region connectivity are the sum of the individual regional percentages.”

**Author response table 1. resptable1:** 

	sDH	dDH	IG	VH
sDH	ü (8)	ü (20)	ü (14)	ü (14)
dDH		ü (12)	ü (18)	ü (18)
IG			ü (6)	ü (12)
VH				ü (6)

If, however, one was only to consider the subset of within-region connections from the broader analysis, the relative representation of sDH, dDH, IG, and VH would be equivalent to that expected if the analysis was *only* performed on within-region connections and used 32 electrodes as its denominator rather than 128. In other words, considering all within- and between-region connections (as reported in the manuscript), the ratio of expected dDH→dDH connections (9.4%) to sDH→sDH connections (6.3%) is 1.5, the same ratio as if the dDH (37.5%) and the sDH (25%) were considered only with the IG and VH and without between-region connections.

Changes made to the manuscript: We have updated the Results section of the manuscript to contain this additional description.

How did the authors verify that the bootstrapped synthetic data converged to the theoretical predictions?

Convergence is verified by computing the within and between-region connection percentages for each simulation run, then taking the mean percentage across all runs. This procedure is performed separately for the synthetic urethane and synthetic isoflurane datasets. These values are illustrated for synthetic urethane in the white bars of Figure 8a. They are then compared to the theoretical predictions (black bars in Figure 8a), which were computed as detailed above. As can be seen in Figure 8, there is very close agreement between the predicted values and simulated data.

5) Lines 495-542: The synthetic data do yield substantially percentages of connectivities, averaging roughly two-thirds the values for the actual data. Furthermore, it is of considerable concern and puzzlement that the synthetic connections for a number of regions are greater than the actual connections (Figure 8.b). What does this mean? For example, Figure 8 appears to show that dorsal and ventral horns are less connected in the actual data than in the supposedly random synthetic data. This seems to contradict the conclusion stated in the abstract that "we…demonstrate that spontaneous functional connectivity also links sensory and motor-dominant regions during unconsciousness."

The reviewers make an important observation, and we apologize that this issue was insufficiently addressed in the Discussion. As we noted previously, we are unable to know the absolute neuron density or relative proportion of neurons across anatomical regions. We are also unable to know whether some functional classes of neurons or anatomically localized groups of neurons are more likely than others to be spontaneously active. Thus, the *primary* utility – and the most direct application – of the synthetic dataset is to estimate whether the proportion of significantly correlated unit pairs observed in the real data is likely to have emerged by chance. And indeed, this was borne out in our original comparisons between observed and synthetic data (with ±5 msec jitter) and in the ±50 msec jitter simulation introduced above.

We feel that there is a fair degree of agnosticism associated with the specific question of whether the observed between-region sensorimotor connectivity is more or less than what would truly be expected at random. Again, the primary reasons being that it is not possible to know what percentage of the total population of spontaneously active neurons our sample represents or to comprehensively estimate what neurons and anatomical regions are most likely to exhibit spontaneous activity. Several other factors also contribute to this uncertainty. In any neural system, one would predict increased synchrony amongst spatially co-localized neurons and less synchrony between spatially distant neurons. Thus, our ability to detect between-region connectivity using individual spike trains is presumably not uniform. The lack of overt stimuli and neural drive in our preparation also suggests that the discharge rate of spontaneously active neurons was likely lower than during awake, behaving conditions, which renders temporal correlation analyses more difficult due to fewer chances to observe coincident spikes. These factors would have contributed to a preferential underestimation of between-region connectivity compared to within-region connectivity and could underlie a portion of the differences we see in the observed vs. synthetic datasets.

What we can say with confidence from the sensorimotor connectivity analyses is that the observed proportions are non-random and non-zero, rejecting the null hypotheses of the analyses. In some sense given the above considerations, we find it surprising that statistically non-zero single-unit connectivity is evident between regions at all; that approximately 1 in every 5 observed connections spanned sensory-dominant and motor-dominant regions seems somewhat remarkable. Nevertheless, we stress caution in interpreting the apparent topological differences between the observed and synthetic datasets.

One potential approach to reconciling the topological differences in our observed and synthetic datasets could be to compute cumulative spike trains (CST) from anatomically co-localized groups of neurons (Negro and Farina, 2012). We could then use CST’s rather than individual neuron spike trains to quantify correlations (or coherence) between regions. CST’s could overcome some of the challenges associated with low discharge rates in individual neurons, more robustly revealing periods of common discharge in a given region and in turn potentially enhancing our ability to detect long-range connectivity (Negro and Farina, 2012). Use of CST’s to infer long-range connectivity has become common in the motor unit field, where coherence between motor unit spike train CSTs is instrumental in inferring cortico-muscular and bulbospinal connectivity, for example (Negro and Farina, 2011; Dideriksen et al., 2018; Thompson et al., 2018). We did not take the CST approach in the present manuscript because we desired to focus specifically on neuron-level firing dynamics, and use of the CST would eliminate that level of granularity. However, CST-based analyses would seem to be an excellent extension of the present work and worth systematically investigating and disseminating in a subsequent manuscript.

Changes made to the manuscript: We have addressed this question in a thoroughly revised and expanded portion of the Discussion section of the manuscript. We elected not to paste the new section here for brevity.

6) The authors attribute the patterns of spontaneous activity found to reflect intrinsic spinal circuit activity, while acknowledging the possibility of sensory afferent feedback contributing to the spontaneous activity despite urethane anesthesia and isoflurane anesthesia in another experimental cohort. It would be important for the authors to discuss whether they have assessed if the spontaneous activity patterns are affected by deeper anesthetic levels than with the standard dose of urethane used for these studies.

We have not tested multiple levels of urethane. As the reviewer notes, we focused on anesthetics with differing mechanisms of action to address the question of dorsal vs. ventral excitability and sensorimotor synchrony. That being said, our experiments do require deep, surgical plane anesthesia, including a lack of withdrawal reflexes. This contrasts with some urethane-anesthetized preparations, in which anesthetic depth is set to *maintain* withdrawal reflexes. Nevertheless, it is an interesting question to consider, particularly considering the observation that patterns of supraspinal functional connectivity appear to vary with sleep stage.

Changes made to the manuscript: We have acknowledged in the Discussion of the manuscript that anesthetic depth could impact the proportion and/or topology of functionally connected units, and that we did not systematically explore this variable.

Also, despite the authors' arguments about some potential disadvantages of deafferentation, this is still an effective way to determine if there are any contributions of local afferent inputs after positioning the microelectrode arrays, particularly since they have confined their electrophysiological recordings to a single lumbar spinal segment and local deafferentation could readily be implemented. Additional information in this regard would be important to strengthen the authors' arguments about the recorded activity reflecting primarily intrinsic spinal circuit activity.

We have conducted additional experiments to address this important issue. The results of these experiments support our interpretation that the intraspinal neural synchrony we observed arises primarily from intrinsic spinal sources and not from latent sensory feedback. The results are summarized in a new figure, included in the manuscript as the new Figure 9.

After achieving our target plane of anesthesia for intraspinal electrode implantation, we exposed the sciatic nerve of the ipsilateral hindlimb proximal to its bifurcation into the tibial and peroneal nerves (Figure 9a, location indicator #1). Using a silver hook electrode, we then recorded both spontaneous and induced neural transmission in the nerve (electroneurogram, ENG). By recording ENG proximal to the bifurcation, we were able to characterize the relative presence or absence of neural transmission as we mechanically probed the L4, L5, and L6 dermatomes. For induced neural transmission, we explored both cutaneous sensory transmission and proprioceptive feedback. We then performed a nerve block using an epineurial lidocaine injection. In this way, we pharmacologically deafferented spinal segments L4-L6 without causing ectopic discharge. Subsequently, we compared the presence and magnitude of induced and spontaneous ENG in the blocked nerve to that observed during our standard anesthetic and experimental regimen. The results of these experiments are as follows:

1. We first note that in the anesthetized, *unblocked* state, spontaneous action potential discharge was not evident (Figure 9b). Overall ENG activity was also negligible during periods of time without sensory stimulation.

2. We then mechanically probed the L4-L6 dermatomes to quantify potential differences in afferent transmission between periods with sensory stimulation and those without. Representative epochs of each type are shown in Figure 9, panels *c* and *d*. In Figure 9c, we show ENG in response to light touch of the L4-L6 dermatomes, including over regions of glabrous and of hairy skin. Dots above the ENG are rasters of individual spikes discriminated from the compound action potential/multi-unit ENG activity. In Figure 9d, we show ENG during periods of induced proprioceptive transmission as we plantarflex and dorsiflex (and hold, as indicated) the ankle. In both panels, the horizontal dashed line indicates the average ENG amplitude during bursts of induced sensory transmission, and the solid horizontal line below it indicates the average ENG amplitude during periods *without* sensory stimulation plus 3x its standard deviation. During epochs between delivery of sensory stimuli, the hindlimb rested gently on a pad with the plantar surface of the hindpaw facing upwards, as during our intraspinal recording sessions (which is also the same as in Figure 9b). Note that no persistent or spontaneously arising action potential discharge is evident during epochs between sensory stimulation; that is, the nerve returns to quiescence.

We then enveloped the multiunit ENG (gray lines over the ENG; 250ms envelope window) and extracted amplitude and variability metrics for periods of spontaneous baseline ENG and induced sensory transmission. Across modalities of induced sensory transmission, bursts of multi-unit ENG had an average peak amplitude 16.7x (±0.82 s.e.m.) greater than the average amplitude of spontaneous baseline ENG (*N*=51 pairs). The average ENG amplitude *across* each period of sensory stimulation (not the average peak amplitude, as previously reported) was 3.09x (±0.17 s.e.m.) greater than the sum of the spontaneous baseline ENG amplitude and its standard deviation.

Next, we blocked afferent transmission via epineurial lidocaine injection. In the top panel of Figure 9e, we show periods of induced sensory transmission and interleaved periods of spontaneous baseline ENG *prior* to nerve block. In the bottom left panel of Figure 9e, we show a representative epoch of neural data recorded 30min after lidocaine injection. No sensory stimulation was delivered during this epoch, and spontaneous action potential discharge expectedly remained absent. In the bottom right panel of Figure 9e we confirm the blocking effect of lidocaine injection by attempting to induce sensory transmission. Qualitatively, it is evident that the spontaneous baseline ENG is indistinguishable pre-and post-block (Figure 9b. vs Figure 9e). This similarity is represented quantitatively in Figure 9f, which shows that the average spontaneous baseline ENG before the block (black bar) is indistinguishable from that during the block (gray bar). Figure 9f also depicts the average peak ENG burst amplitude across trials (white bar) to provide reference for the observed spontaneous baseline ENG amplitude.

Together, these results demonstrate that latent afferent transmission is negligible in our preparation and is therefore unlikely to contribute substantially to the spontaneous intraspinal neural transmission we observed. By extension, this finding suggests that neurons in intrinsic spinal circuits themselves are presumably an integral component of the activity.

Changes made to the manuscript: Changes related to this question are numerous and are thus only summarized here. Please refer to the locations indicated below in the manuscript’s main body for additional details. (1) We have updated the Methods section to describe the new experiments; (2) we have updated the Results section to include Figure 9 and accompanying description of results; (3) we have clarified our discussion of the potential role of afferent feedback in the Discussion section; and (4) the References list has also been updated accordingly.

7) Unless motoneurons are completely inactive under the anesthesia, might high-amplitude spikes in VH identify likely motoneurons? If that is the case, it would be very interesting to assess motoneuron connectivities with other areas and neuronal populations. This deserves discussion.

This is a very interesting question, and one we are actively pursuing in the lab from several angles. Strictly speaking, however, we cannot confirm unequivocally from these data whether a portion of the spikes we observed were from motoneurons. That is because we did not routinely implant EMG leads in these animals and attempt to stimulate across the electrode array to look at output effects (that is part of our ongoing work, however). We suspect that it was unlikely that many of the spontaneously active neurons many were motoneurons, however, because a requirement of our surgical protocol was an absence of withdrawal reflexes.

Although admittedly speculative, if a portion of the spontaneously active units were in fact motoneurons, it could lend support to the hypothesis that central pattern generators may contain latent spontaneous activity in the adult spinal cord. It could also imply that the spinal cord maintains an intrinsic state of readiness to execute sensorimotor behaviors by keeping minimally active a certain proportion of neurons known to be functionally relevant to many spinally-mediated behaviors, particularly avoidance reflexes. This latter topic is the subject of a separate paper we currently have under review, in which we electrophysiologically and phenomenologically classify spontaneously active neurons in a similar experimental preparation.

Please note that while we agree that this topic is ripe for investigation and discussion in the field, we did not amend the Discussion section of this manuscript to include additional details. We feel that any information we provide at this stage would be too speculative to enrich the discussion in a robust manner. We are eager to publish our forthcoming work more directly related to this topic, however, which will support a more thorough discussion of the topic.

References:

Amarasingham A, Harrison MT, Hatsopoulos NG, Geman S (2012) Conditional modeling and the jitter method of spike resampling. Journal of Neurophysiology.

Dideriksen JL, Negro F, Falla D, Kristensen SR, Mrachacz-Kersting N, Farina D (2018) Coherence of the Surface EMG and Common Synaptic Input to Motor Neurons. Frontiers in Human Neuroscience 12.

Fujisawa S, Amarasingham A, Harrison MT, Buzsáki G (2008) Behavior-dependent short-term assembly dynamics in the medial prefrontal cortex. Nature Neuroscience.

Negro F, Farina D (2011) Linear transmission of cortical oscillations to the neural drive to muscles is mediated by common projections to populations of motoneurons in humans. Journal of Physiology 589.

Negro F, Farina D (2012) Factors Influencing the Estimates of Correlation between Motor Unit Activities in Humans. PLoS ONE 7.

Pedreira C, Martinez J, Ison MJ, Quian Quiroga R (2012) How many neurons can we see with current spike sorting algorithms? Journal of Neuroscience Methods.

Quiroga RQ, Nadasdy Z, Ben-Shaul Y (2004) Unsupervised spike detection and sorting with wavelets and superparamagnetic clustering. Neural Computation.

Rey HG, Pedreira C, Quian Quiroga R (2015) Past, present and future of spike sorting techniques. Brain Research Bulletin.

Thompson CK, Negro F, Johnson MD, Holmes MR, McPherson LM, Powers RK, Farina D, Heckman CJ (2018) Robust and accurate decoding of motoneuron behaviour and prediction of the resulting force output. Journal of Physiology 596.